**Data Availability Statement:** Authors uploaded datasets used in the study on climate data (both observed and selected GCMs), survey data on crop

# Simulating adaptation strategies to offset potential impacts of climate variability and change on maize yields in Embu County, Kenya

Sridhar Gummadi[1]*, M. D. M. Kadiyala[2], K. P. C. Rao[1], Ioannis Athanasiadis[3], Richard Mulwa[4], Mary Kilavi[5], Gizachew Legesse[1], Tilahun Amede[1]

**1** International Crops Research Institute for the Semi-Arid Tropics (ICRISAT), Addis Ababa, Ethiopia, **2** Acharya N. G. Ranga Agricultural University (ANGRAU), Guntur, Andhra Pradesh, India, **3** Wageningen University, Gelderland, The Netherlands, **4** Centre for Advanced Studies in Environmental Law and policy (CASELAP), University of Nairobi, Nairobi, Kenya, **5** Kenya Meteorological Department, Nairobi, Kenya

* sridhar.gummadi@irri.org

## Abstract

In this study, we assessed the possible impacts of climate variability and change on growth and performance of maize using multi-climate, multi-crop model approaches built on Agricultural Model Intercomparison and Improvement Project (AgMIP) protocols in five different agro-ecological zones (AEZs) of Embu County in Kenya and under different management systems. Adaptation strategies were developed that are locally relevant by identifying a set of technologies that help to offset potential impacts of climate change on maize yields. Impacts and adaptation options were evaluated using projections by 20 Coupled Model Intercomparison Project—Phase 5 (CMIP5) climate models under two representative concentration pathways (RCPs) 4.5 and 8.5. Two widely used crop simulation models, Agricultural Production Systems Simulator (APSIM) and Decision Support System for Agrotechnology Transfer (DSSAT) was used to simulate the potential impacts of climate change on maize. Results showed that 20 CMIP5 models are consistent in their projections of increased surface temperatures with different magnitude. Projections by HadGEM2-CC, HadGEM2-ES, and MIROC-ESM tend to be higher than the rest of 17 CMIP5 climate models under both emission scenarios. The projected increase in minimum temperature (Tmin) which ranged between 2.7 and 5.8°C is higher than the increase in maximum temperature (Tmax) that varied between 2.2 and 4.8°C by end century under RCP 8.5. Future projections in rainfall are less certain with high variability projections by GFDL-ESM2G, MIROC5, and NorESM1-M suggest 8 to 25% decline in rainfall, while CanESM2, IPSL-CM5A-MR and BNU-ESM suggested more than 85% increase in rainfall under RCP 8.5 by end of 21st century. Impacts of current and future climatic conditions on maize yields varied depending on the AEZs, soil type, crop management and climate change scenario. Impacts are largely negative in the low potential AEZs such as Lower Midlands (LM4 and LM5) compared with the high potential AEZs Upper Midlands (UM2 and UM3). However, impacts of climate change are largely positive across all AEZs and management conditions when $CO_2$ fertilization is included. Using the differential impacts of climate change, a strategy to adapt maize

cultivar, soils, crop management and crop simulated yields for the baseline, future climate with and without adaptation. Historical climate data for the Embu county are obtained from Kenya Meteorological Department (KMD) https://www.meteo.go.ke/, data policy on use of climate records need to be acknowledged. Data can be accessed from: https://doi.org/10.7910/DVN/0ECMP0 (Crop Simulation models (APSIM and DSSAT Calibration); https://doi.org/10.7910/DVN/QLSDSW (Embu Climate data both historical and future projections from 20 AOGCMs); https://doi.org/10.7910/DVN/DVAO17 (Maize crop simulations using APSIM and DSSAT at 4 locations in Embu county Kenya for the current and future climates); https://doi.org/10.7910/DVN/EWBQGD (Possible adaptation options for maize crop in Embu county Kenya in the future climate projections).

**Funding:** This work was implemented as part of the CGIAR Research Program on Climate Change, Agriculture and Food Security (CCAFS), which is carried out with support from CGIAR Fund Donors and through bilateral funding agreements. For details please visit https://ccafs.cgiar.org/donors. The Agricultural Model Intercomparison and Improvement Project (AgMIP) - The future of food and farming in Sub-Saharan Africa was funded by the Department of International Development, United States Department of Agriculture and implemented by Columbia University and International Crops Research Institute for the Semi- Arid Tropics. AgMIP project provided support in the form of salary for authors KPCR and SG during the period 2013-2015. AgMIP global team leaders developed the protocols for data collection and analysis, but the funders had no role in study design, data collection and analysis, decision to publish, or preparation of the manuscript.

**Competing interests:** The authors whose names are listed immediately below certify that they have NO affiliations with or involvement in any organization or entity with any financial interest (such as honoraria; educational grants; participation in speakers' bureaus; membership, employment, consultancies, stock ownership, or other equity interest; and expert testimony or patent-licensing arrangements), or non-financial interest (such as personal or professional relationships, affiliations, knowledge or beliefs) in the subject matter or materials discussed in this manuscript. Author names: Sridhar Gummadi Kadiyala M.D.M K. P. C. Rao Ioannis N. Athanasiadis Richard Mulwa Mary Kilavi and Gizachew Legesse.

cultivation to climate change in all the five AEZs was identified by consolidating those practices that contributed to increased yields under climate change. We consider this approach as more appropriate to identify operational adaptation strategies using readily available technologies that contribute positively under both current and future climatic conditions. This approach when adopted in strategic manner will also contribute to further strengthen the development of adaptation strategies at national and local levels. The methods and tools validated and applied in this assessment allowed estimating possible impacts of climate change and adaptation strategies which can provide valuable insights and guidance for adaptation planning.

## 1. Introduction

Agriculture is and will continue to be the main livelihood for millions of smallholder farmers in Africa and other developing countries across the world. In Eastern Africa, agriculture accounts for 43% of GDP and contributes to more than 80% employment [1]. The region experiences high variability in rainfall [2,3] which has a direct bearing on the performance of agriculture. Generally, the region experiences prolonged and highly destructive droughts covering large areas at least once every decade and more localized events almost every year [4,5]. In the countries such as Ethiopia, where agriculture is the main driver of the economy, the economic activity measured by the gross domestic product is closely linked to the variability in rainfall [6]. According to [7], a single drought event in a 12-year period reduces GDP by 7–10% and increase poverty by 12–14% in Eastern Africa.

In Africa in general and Eastern Africa in particular, agriculture is predominantly rainfed [8] and the production is, therefore, heavily influenced by various climate dependent biotic and abiotic factors. Important among them is the plant available water, the dynamics of which are directly associated with rainfall and the spatial and temporal variability in it. The impacts of climate variability will be more severe on rainfed systems of the semi-arid tropics which have a marginal environment for crop production and adding to this, climate change is expected to further exacerbate these challenges. The fifth assessment report (AR5) of IPCC concluded with high certainty that the global climate change is unequivocal and will continue for the next few decades even if the greenhouse gas emissions are contained at the current level [9]. Coping with the potential impacts from projected changes in climate on agriculture is high on the agenda of most African countries which are currently struggling to meet the increasing demand for food and income from the rapidly growing population.

With nearly 80% of the land area under arid and semi-arid environments [10], Kenya is one of the highly vulnerable countries to climate change. A number of studies suggest that maize yields in Kenya are going to be negatively impacted at the national level [11,12]. However, the reliability of such impact studies is highly dependent on the skill of the simulation models used and the underlying assumptions in setting the model scenarios [13]. In assessing the impacts of climate change, skills of both climate models that are used to generate future climate scenarios and crop simulation models used to evaluate the impacts of projected climate conditions on crop growth are important. Though there has been a significant progress in modelling climate processes, there are still major issues due to the variety of spatial scales used and the bias associated within the climate models such as internal variability, inadequate parameterization, model and scenario uncertainty [14,15]. A wide range of crop models ranging from very detailed process models to the relatively simple statistical models were applied to

assess the impacts of changing climate on crop production [16]. To date, much of the available information on impacts of climate change is at scales much bigger than the farm and is based on simple statistical or rule-based models which will not be of much help in identifying the components of the system that are vulnerable to climate change. For designing effective adaptation strategies at farm level, information about the climate sensitivity of various crops and management practices employed is an essential pre-requisite. Such detailed assessment of climate sensitivity of components of agricultural systems is possible with process-based system simulation models such as APSIM and DSSAT but the data required to calibrate, validate and run the models is not readily available.

The climate change information required for conducting impact assessments using the crop simulation models APSIM and DSSAT is of a spatial scale much finer than that provided by GCMs [17–21]. The GCMs have resolution of hundreds of kilometres perhaps as coarse as 300 x 300 km, while Regional Circulation Models (RCMs) are fine-scale of tens of kilometres (50x 50 km). However, many impact applications require the equivalent of point climate or station observations and are highly sensitive to coarse-scale climate scenarios generated by GCMs. This is particularly true for regions of complex topography, such as Kenya. The most straightforward means of resolving the spatial scale issues is to downscale GCMs climate projections to finer-scales. Two different approaches, dynamical and statistical were employed in downscaling coarse GCM projections to local applications [22]. Since dynamical downscaling has similar uncertainty issues to GCMs and computationally intensive, statistical downscaling methods are more commonly used to generate climate change projections at point or station level because of its relative ease of use and lower time, data and resource requirement [23].

Various system simulation models are being used to make detailed assessment of the impacts of climate change on various components of the smallholder farming systems such as crops, cultivar and management options. APSIM and DSSAT are the two models that are most widely used in assessing the climate impacts on agricultural systems. When properly calibrated and validated, these models can simulate the growth and performance of a wide range of crops as a function of climate, soil and crop management [24,25]. However, these models are data intensive and require careful calibration and validation using site and location specific climate, soil and crop management information. AgMIP developed a set of protocols [26] that integrate state of the art climate, crop and economic simulation models, at different time-scales and under different emissions scenarios for a more comprehensive assessment of climate change impacts. The methodology involves development of downscaled future climate scenarios using the simple delta approach and assessing the impacts of current and future climatic conditions on agricultural systems using process-based crop simulation models such as APSIM and DSSAT. The objective of this study is to make a detailed assessment of climate change impacts on agricultural systems in Embu County, Kenya using AgMIP developed protocols and identify potential options for adaptation. More specifically, the study is aimed at:

a. Assessing current variability and projected changes in rainfall, Tmax and Tmin by downscaling and analysing location specific climate change scenarios to mid and end century periods under RCPs 4.5 and 8.5 for various AEZs, namely, Upper Midlands (UM2, UM3) and Lower Midlands (LM3, LM4 and LM5) of Embu county in Kenya.

b. Assessing the impacts of climate variability and change on maize yields in different AEZs of Embu county and identify key vulnerabilities to climate factors.

c. Identifying and evaluating adaptation options that make maize production in the Embu county more resilient to current and future climatic conditions.

## 2. Materials and methods

### 2.1 Study location

The study was conducted in Embu County in Kenya, which is characterised by a range AEZs ranging from highlands with altitudes up to 4,500 m in the North West which is part of Mt Kenya to lowlands with altitudes around 500 m in the East in the Tana River basin. The climate of the county fluctuates in accordance with the altitudinal variations. The average annual rainfall varies from more than 2200 mm at an altitude of 2500 m to less than 600 mm at an altitude of 700 m near Tana River, while temperature varies from 20˚C to 30˚C. July is usually the coldest month with the average monthly temperature of 15˚C while September is the warmest month with an average monthly temperature rising up to 27.1˚C [27]. The county is characterized by bimodal rainfall pattern with two distinct rainy seasons. The two seasons are generally referred to as Long Rains (LR) season occurring between March and May and Short Rains (SR) season between October-December. In Embu, cropping is done in both LR and SR seasons. Though both seasons receive similar amounts of rainfall, SR season with slightly higher rainfall and longer growing season is generally considered as more dependable.

The county was selected based on its representativeness of the country's major AEZs and based on the availability of the data (crop, soil and climate) required to parameterize the crop simulation models. The county is divided into 11 AEZs based on their probability of meeting the temperature and water requirements of the major crops grown in the country. Among them, Upper Midlands (UM2, UM3 and UM4) and Lower Midlands (LM3, LM4 and LM5) are the major AEZs that represent the main cropping areas in Embu County (Fig 1). The other AEZs, representing upper highland (UH0), lower highland (LH0 and LH1) and inner lowland (IL5), are either too cool and wet or too hot and dry for crop production and hence excluded from this analysis. Tea and coffee dominate the highland cropping systems. The AEZs of, UM2, UM3 and LM3 with an annual rainfall of more than 1000 mm are generally considered as high potential agricultural areas, while the AEZs, LM4 and LM5 with less than 1000 mm rain are considered as low potential agricultural areas. This analysis is limited to these five AEZs where the main food crop maize is extensively grown. Though maize is a common crop in all the five AEZs, there are marked differences between the AEZs in the way the crop is managed. For example, farmers in the high potential areas favour long duration maize cultivars with relatively higher application of inputs such as fertilizer while those in the low potential areas favour short duration maize cultivars with low levels of input use.

### 2.2 Current climate variability and future climate conditions

Long term observed climate data for the baseline period 1980–2010 was collected from the archives of Kenya Meteorological Department for 4 stations (Embu, Karurumo, Ishiara and Kindaruma) that are located within the target areas as presented in Fig 1. All stations have long-term (30 years) rainfall data with less than 10% missing data. Good quality temperature data is available for Embu station which is one of the synoptic stations managed by Kenya Meteorological Service. The data was subjected to quality checks to identify outliers, typos and discontinuity errors using R-Climdex [28] and where necessary AgMERRA Climate Forcing Dataset for Agricultural Modelling [29] data was used to fill missing data and replace the outliers in the observed data.

In this study the non-parametric Mann-Kendall trend test, a widely used statistical test for the analysis of trend in climatologic and hydrological studies was used [30,31]. This method has two advantages, first, it is a non-parametric test and does not require the data to be

## I.     List of figures

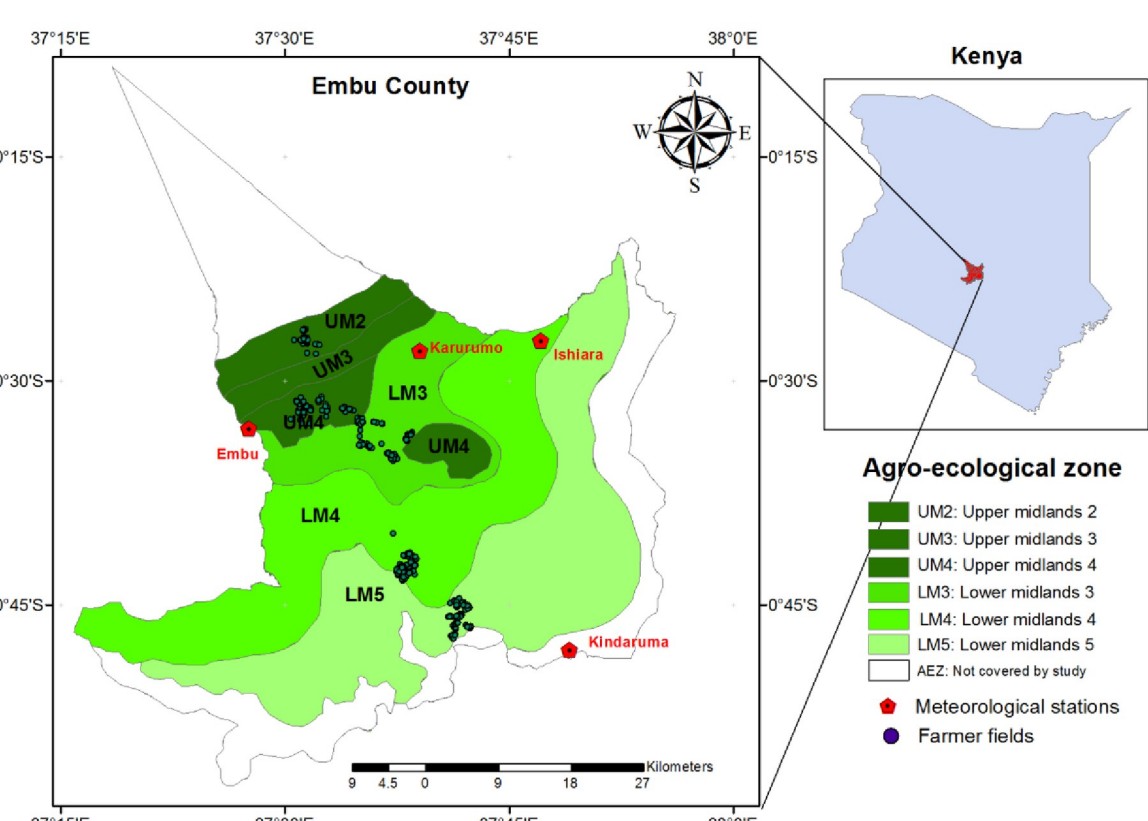

**Fig 1. Agro-ecological zones of Embu County in Kenya (inset) with locations of the meteorological stations and locations of the farmers covered by the survey.**

normally distributed. Second, the test has low sensitivity to abrupt breaks due to heterogeneous time series [32]. In this method, a correlation coefficient, tau, is computed, which has a value between -1 and 1 and denotes the relative strength of the trend in a time series.

Location specific climate change scenarios were developed using a statistical downscaling technique. The method used is known generally as delta method in which absolute monthly changes in both Tmax and Tmin and relative changes in precipitation were computed and these changes are perturbed to the corresponding observed historical variables [33]. The delta method assumes that future model biases for both mean and variability will be the same as those in present day simulations. Delta method calculates changes in surface temperatures ($\Delta$T) (Eq (1)) and precipitation ($\Delta$P) (Eq (2)) and perturb the projected changes to observed climate data as shown in Eqs 3 and 4. Surface temperatures are adjusted by adding the difference obtained from Eq 1. The daily precipitation is adjusted by multiplying the precipitation ratio (Eq 4). The method assumes that changes in climates are only relevant at coarse scales and that relationships between variables are maintained towards the future. This method was applied over 20 CMIP5 GCMs model groups involved to run their models for future conditions of greenhouse gas emissions, so-called RCPs. A set of four different pathways are [34] defined based on the radiative forcing in W m$^{-2}$ at the end of the 21$^{st}$ century as RCP2.6,

RCP4.5, RCP6.0, to RCP8.5 [35,36].

$$\Delta T = (\bar{T}_{fut} - \bar{T}_{base}) \tag{1}$$

$$\Delta P = (\bar{P}_{fut}/\bar{P}_{base}) \tag{2}$$

$$T_{adj} = \bar{T}_{obs} + \Delta T \tag{3}$$

$$P_{adj} = \bar{P}_{obs} \, x \, \Delta P \tag{4}$$

Where

$\bar{T}_{fut}$: Climate model future projected temperatures

$\bar{T}_{base}$: Climate model simulated baseline temperatures

$\bar{P}_{fut}$: Climate model future projected precipitation

$\bar{P}_{base}$: Climate model simulated baseline precipitation

$\bar{T}_{obs}$: Historical surface temperature

$\bar{P}_{obs}$: Historical precipitation

Twenty CMIP5 GCMs were selected for the current study to cover the full spectrum of projections in future precipitation and surface temperatures. Since GCMs differ in their projections because of differences in underlying assumptions and the way climate system processes are simulated, IPCC launched the CMIP5, whereby a multi-GCM ensemble analysis was facilitated through the provision of climate model outputs. Due to high sensitivity of agriculture to variability in climatic conditions, the differences in the projections by different GCMs are expected to have differential impacts. These uncertainties arising from climate change projections are handled by comparing the performance of maize yields simulated with outputs from different GCMs that IPCC included in the CMIP5 assessments. This helped in understanding the impacts of wise range of projected climates on maize yields and identify robust adaptation options. In this study we deployed projections from 20 GCMs under two RCPs for two different time periods. Climate change scenarios were developed for mid-century (2041–2070) and end-century (2071–2100) for two RCPs. Selected RCPs (4.5 and 8.5) represent the realistic and pessimistic emission scenarios. We used 20 GCMs in this study since the usage of multiple models was suggested to provide more reliable assessment of impacts of climate change on weather sensitive sectors like agriculture [26,37,38]. Climate change impact assessment studies particularly, agricultural systems which are sensitive to climate variability and change show that the agricultural sector is aversively affected and the situation is expected to worsen in the future [26]. Cropping systems impact studies of climate change should not be assessed using only one GCM, as major source of uncertainty for projections of future climate are from unknown future trajectories of $CO_2$ and $CH_4$, emissions, but also due to the highly simplified representation of reality encoded in these models [39]. The use of multiple models can provide more reliable decision support in climate change impact assessment and assessments of agricultural system vulnerability [26,37,38]. The $CO_2$ concentration are adjusted to correspond the RCP and time periods defined in [40] for regional assessment. The concentrations used are 499 ppm to mid-century, 532 ppm to end-century under RCP 4.5 and 571 ppm to mid-century and 801 ppm to end-century under RCP 8.5. R scripts developed by AgMIP climate team were used to generate required climate scenarios [29].

## 2.3 Crop simulation models (CSMs)

In this study we used two plot specific widely applied CSMs—APSIM [25] version 7.7 and DSSAT version 4.7, CERES-Maize [24,41] to assess the performance of maize under different climatic conditions. These are process based models operating on a daily time step with a capability to dynamically simulate the main processes of crop growth and development, such as phenological development, biomass production and grain yield as a function of climate, soil, crop and management. CSMs simulate phenological development of the crop based on accumulated thermal time derived from the daily surface temperatures (Tmax and Tmin) and biomass development based on radiation use efficiency (RUE). Biomass partitioning rates to different plant parts vary with crop development stage and re-translocation begins at the stage of starting grain filling. These models have been evaluated around the globe under different soil, climate and management conditions [42,43] and the simulated yields were found to be realistic and reliable measures of actual yields. One of the limitations in using CSMs is the amount of data required to define the soil, crop and management variables. Extensive efforts were made to compile the required data from relevant secondary sources that included formal and informal publications and by conducting household surveys.

## 2.4 Model parameterization

**2.4.1 Farm and farm management data.** Since management varies form one farmer to the other, household surveys covering a total of 440 households were conducted in 2013 to capture diversity and variability in the resources and management of maize production systems in the target AEZs. The households for the survey were identified using a combination of stratified and multistage sampling technique (Table 1). The survey was conducted by University of Nairobi and Kenya Agricultural Research Institute (KARI) using the protocols developed by ICRISAT and detailed description of the methodology used and other survey details are in [44]. Briefly, in all the five selected AEZs one sub-location (In Kenya sub-location is the fifth level administrative division after province, district, division and location under its old constitution) was chosen for sampling (Fig 1). At the selected sub-location, data on household size, farm size, soil type, crops grown, management practices employed, yields achieved and sources of income was collected.

**2.4.2 Soil data.** Soil data were obtained from the soil survey reports of KARI by identifying representative soil profiles for the selected AEZs. Soil profile data as required for CSMs (APSIM and DSSAT) were created for each soil types identified using the data from the benchmark soil profile. To account high variability in the soil conditions across the farms, two variants, one representing the good and the other representing poor were created by increasing or decreasing the soil organic matter and plant available water contents by 20%. These profiles are then assigned to individual farms based on the location of the farm and perception of the farmer about the fertility status of the farm captured during the household survey. During the

**Table 1. Agro-ecological zone (AEZ) wise number of households covered by the household survey and the administrative divisions they belong to in Embu county.**

| AEZ | Division | Number of HHs |
|---|---|---|
| Upper Midland 2 | Kevote, Nembure | 73 |
| Upper Midland 3 | Kithimu, Nembure | 87 |
| Lower Midland 3 | Riandu, Siakago | 107 |
| Lower Midland 4 | Nyangwa, Gachoka | 91 |
| Lower Midland 5 | Mavuria, Gachoka | 82 |

**Table 2. Important characteristics of the representative soil profiles used with crop simulation models and the agro-ecological zones they represent.**

| Properties | Embu | Kavutiri | Gachuka | Machanga |
|---|---|---|---|---|
| Target Agro-ecology | UM2 | UM3 and LM3 | LM4 | LM5 |
| Soil type | Typic Palehumult | Othoxic Palehumult | Typic Haplorthox | Xanthic Ferralsol |
| Soil layers/depth (cm) | 4/102 | 6/200 | 4/104 | 4/80 |
| Sand, silt, clay (% in 0-15cm) | 20,24,56 | 20,26,54 | 20,24,56 | 66,12,22 |
| Plant available water (mm) | 93.7 | 152.2 | 89.4 | 100 |
| Organic matter (% in top three layers) | 2.09, 1.49, 0.91 | 3.61,2.29,1.58 | 2.29, 1.58,0.92 | 0.58, 0.50,0.40 |

survey, farmers were asked to rate fertility status of their farm as good, average and poor when compared to general conditions in that area. This information was used to identify appropriate soil profile for individual farmers. A total of 12 soil profiles were developed with all the required parameters (Table 2) for CSMs APSIM and DSSAT. Key characteristics of the average soil profiles used with the crop models are as presented in Table 2.

**2.4.3 Crop data.** Much of the crop data required is for developing maize cultivar specific parameters to capture the differences between the maize cultivars in phenological development and yield potential. We calibrated APSIM and DSSAT for three varieties that represent the long, medium and short maturity types with different yield potential using the experimental data from a study conducted on the Embu research farm of Kenya Agricultural Research Institute (Table 3). The varieties selected are H513 for long duration, H511 for medium duration and Katumani for short duration. The experiment included all these varieties and was conducted over three seasons i.e., SR season of 2000 and LR and SR seasons of 2001. All the available data was compiled from the research reports as well as personnel communication with the researcher concerned. This included, dates of sowing, days to tasselling and flowering, days to maturity and grain and dry matter yields at harvest. Data on biomass at different days after sowing was available for some seasons.

**2.4.4 Model calibration and validation.** The calibration and validation process can determine to what extent CSMs can reproduce experimental observations, such as crop phenology and yield components. Cultivar specific parameters were derived by adjusting the thermal

**Table 3. Maize varieties used by farmers and the identified equivalent in the model.**

| Variety used by farmer | Duration (Months) | Yields (t/ha) | Variety in the Model |
|---|---|---|---|
| DK41 | 3.5 to 4.5 | 5–6 | Deka_lb |
| DK43 | 4–5 | 6–7 | H511 |
| H513 | 4–5 | 6–8 | H511 |
| H613 | 6–8 | 8–10 | H513 |
| Local | All | 3–5 | Katumani |
| Duma | 4–5 | 6–7 | H511 |
| Pioneer | 5–6 | 8–10 | H513 |
| Variety used by farmer | Duration (Months) | Yields (t/ha) | Variety in the Model |
| DK41 | 3.5 to 4.5 | 5–6 | Deka_lb |
| DK43 | 4–5 | 6–7 | H511 |
| H513 | 4–5 | 6–8 | H511 |
| H613 | 6–8 | 8–10 | H513 |
| Local | All | 3–5 | Katumani |
| Duma | 4–5 | 6–7 | H511 |
| Pioneer | 5–6 | 8–10 | H513 |

**Table 4. Genetic coefficients for three maize varieties derived from the calibration with APSIM and DSSAT using experimental data from Embu, Kenya.**

| | DSSAT | | | | | |
|---|---|---|---|---|---|---|
| **CULTIVAR** | **P1** | **P2** | **P5** | **G2** | **G3** | **PHINT** |
| **KATUMANI** | 100.0 | 0.500 | 554.0 | 550.0 | 10.60 | 47.0 |
| **H511** | 190.0 | 0.600 | 725.0 | 550.0 | 7.90 | 42.0 |
| **H513** | 205.0 | 0.600 | 760.5 | 690.0 | 8.70 | 40.0 |
| | APSIM | | | | | |
| **KATUMANI** | 150 | 24.0 | 660 | 450 | 8.5 | NA |
| **H511** | 180 | 24 | 780 | 650 | 8.0 | NA |
| **H513** | 240 | 20.0 | 980 | 750 | 8.0 | NA |

**P1:** Thermal time from seedling emergence to the end of the juvenile phase (expressed in degree days above a base temperature (8˚C) during which the plant is not responsive to changes in photoperiod.

**P2:** Extent to which development (expressed as days) is delayed for each hour increase in photoperiod above the longest photoperiod at which development proceeds at a maximum rate (which is considered to be 12.5 hours).

**P5:** Thermal time from silking to physiological maturity (expressed in degree days above a base temperature.

**G2:** Maximum possible number of kernels per plant.

**G3:** Kernel filling rate during the linear grain filling stage and under optimum conditions (mg/day).

**PHINT:** Phylochron interval; the interval in thermal time (degree days) between successive leaf tip appearances.

time required to complete various growth stages until the simulated phenology matched the observed phenology. After matching the phenology, adjustments were made to match the simulated biomass and yield with experimental yield. The final set of cultivar specific parameters used in APSIM and DSSAT are summarized in Table 4. After calibration, the models were validated by simulating yields from 160 of the 440 farmers covered by the survey and fall under Embu climate by setting up farmer specific climate, soil, crop and management parameters. The validation is limited to 160 farmers since this is the only location for which climate data for the survey year is available. The simulated yields by both APSIM and DSSAT are found to be generally higher than the yields reported by farmers (Fig 2).

**2.4.5 Crop simulations.** Crop simulations were carried out by setting up simulations with farmer specific soil, crop and management data. A total of 440 farmer fields were simulated

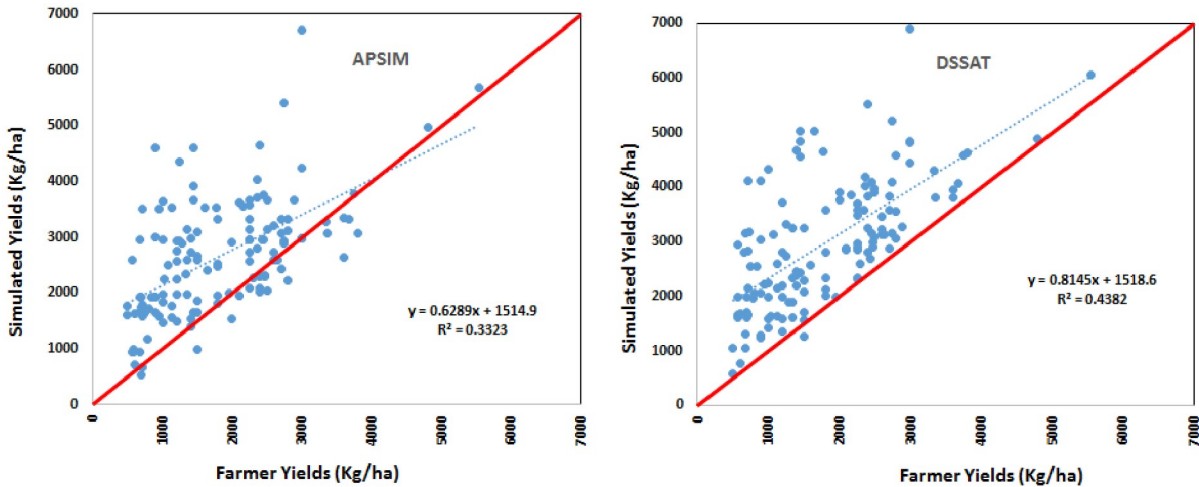

**Fig 2. Relationship between maize yields reported by farmers and simulated by APSIM (left) and DSSAT (right).** Red solid line represents 1:1.

with each of the climate scenarios that included one observed and 80 climate change scenarios. The model simulations were initiated by defining initial conditions that normally exist at the beginning of the season. To account for the biomass leftover from the previous season crop, weeds and other plant material accumulated during the off season, the amount of residue at the beginning of the season was set to 400 kg/ha with a nitrogen content of 0.8%. Inorganic nitrogen in the profile at the beginning of the season was estimated to be 8 kg/ha, 0.1 ppm of $NO_3$ and 0.01 ppm of $NH_4$. Every year the model run was initiated 15 days before the start of planting window and initial moisture was set to 50% of the total available water distributed through the profile. For other parameters we used farmer specific information collected during the survey. This includes information on maize cultivar, planting time, plant population or seed rate and amount of fertilizer and manure applied. The varieties grown by the farmers in the target AEZs were grouped into four groups based on the duration and yield potential of the varieties. We have identified H511, H513, Deka_lb and Katumani varieties to represent these groups. These are also the varieties the CSMs were calibrated and validated using the experimental data. For Deka_lb we used the default parameters from the model. Table 3 presents the farmer used maize cultivar and its equivalent in the CSMs. The results were analysed to identify the climate sensitivity of the systems under different combinations of soil, crop and management conditions.

**2.4.6 Crop management strategies for adaptation.** Potential adaptation options vary with the type and intensity of projected impacts on performance of maize under different climate change scenarios. Since current farmer yields are very low due to low input agriculture practiced by majority of smallholder farmers, it is hypothesized that substantial improvement in maize yields can be achieved with available technologies even under the projected changes in climatic conditions. Adoption of better performing crop varieties with improved crop management practices were evaluated for their ability to cope with increased temperatures and associated changes in rainfall. The adaptation strategy formulated included changes to maize cultivars, planting time, amount of fertilizer applied and plant density. Using crop simulation models, optimal combination of these management practices for each AEZ were identified by conducting a series of simulations using the two crop simulation models.

## 3. Results

### 3.1 Trends in observed climatic conditions

To test the significance of the observed trends in both surface temperatures and rainfall (Table 5) the Mann-Kendall Tau-b non-parametric function is employed. The p-values from the test indicate that the trends in temperature is significant at less than 0.02 level while, the trend in rainfall is less conclusive, except for Kindaruma. Where trends in annual and LR season rainfall has shown a significant increasing trend with p-values of 0.01 and 0.05 respectively.

Analysis of historical climate data at four locations revealed clear increasing trend in both Tmax and Tmin during LR and SR seasons (Fig 3). An increase of 0.54˚C was observed in the annual mean Tmax during the period 2001–2010 compared to that during the period 1981-1990. The corresponding increase in Tmin is 0.3˚C. Compared to 1981–1990 period, the average annual temperatures are higher by 0.57˚C during the SR season and by 0.49˚C during the LR season during the period 2001–2010.

In case of rainfall, no clear declining or increasing trend was observed in the amount of rainfall received annually or during the LR and SR seasons at all the four locations. However, there are indications that variability in rainfall, particularly during the SR season was increasing (Fig 4). The ten-year moving Coefficient of Variation (CV) of SR season rainfall has

**Table 5. Kendall Tau significance test for annual and seasonal temperature at Embu and rainfall at all the four locations.**

| | | Average Temperature | Rainfall | | | |
|---|---|---|---|---|---|---|
| | | Embu | Embu | Ishiara | Karurumo | Kindaruma |
| Annual | Kendall's tau | 0.43 | 0.22 | 0.34 | 0.12 | 0.38 |
| | P-Value | 0.00 | 0.13 | 0.02 | 0.45 | 0.01 |
| Short Rain season | Kendall's tau | 0.30 | 0.26 | 0.14 | 0.14 | -0.05 |
| | P-Value | 0.02 | 0.08 | 0.36 | 0.34 | 0.75 |
| Long Rain season | Kendall's tau | 0.35 | 0.52 | 0.10 | -0.03 | 0.29 |
| | P-Value | 0.01 | 0.00 | 0.51 | 0.86 | 0.05 |

increased from 30% in 1980s to 50% during 2001–2010 period. This is a substantial change from the current situation and is expected to have major impacts on the productivity of smallholder farms since SR season is an important season during which the main food crop maize is extensively grown.

### 3.2 Projected changes in climate conditions

In this paper, we investigated the changes in Tmin, Tmax and rainfall over Embu county under RCP4.5 and RCP8.5. Downscaled future climate projections for different climate change scenarios showed a general increase in surface temperatures at all the four locations in Embu County (Fig 5). The magnitude of this increase over different time periods varied with GCMs and emission scenario that drives the climate models. In general, Tmin and Tmax projections by HadGEM2-CC, HadGEM2-ES, IPSL-CM5A-MR, IPSL-CM5A-LR and MIROC-ESM were found to be higher compared to other selected 17 CMIP5 GCMs. Projected increase in Tmax to end century period by different GCMs varied from 2.02˚C to 4.80˚C and that in Tmin varied from 2.70˚C to 5.80˚C under RCP 8.5 compared with baseline climate. The magnitude of

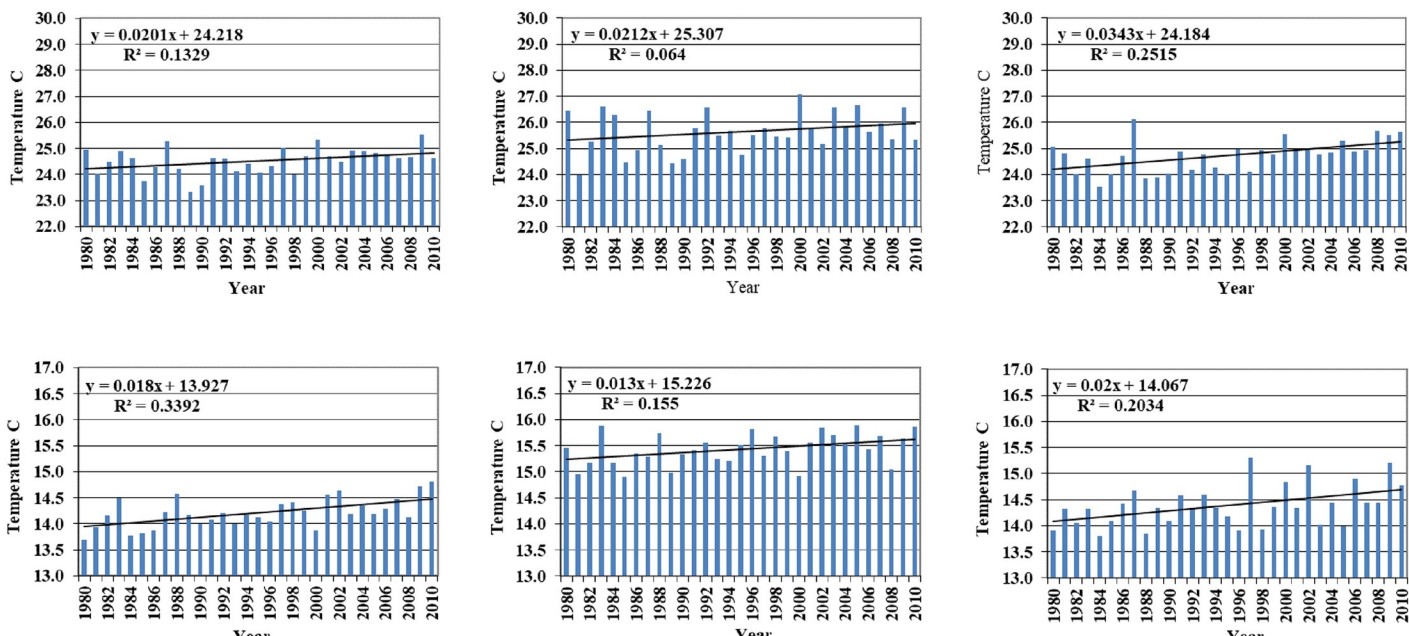

**Fig 3. Trends in annual, long rain (LR)-Season and short rain (SR)- maximum temperature (top) and minimum temperature (bottom) at Embu, Kenya with linear trend line.**

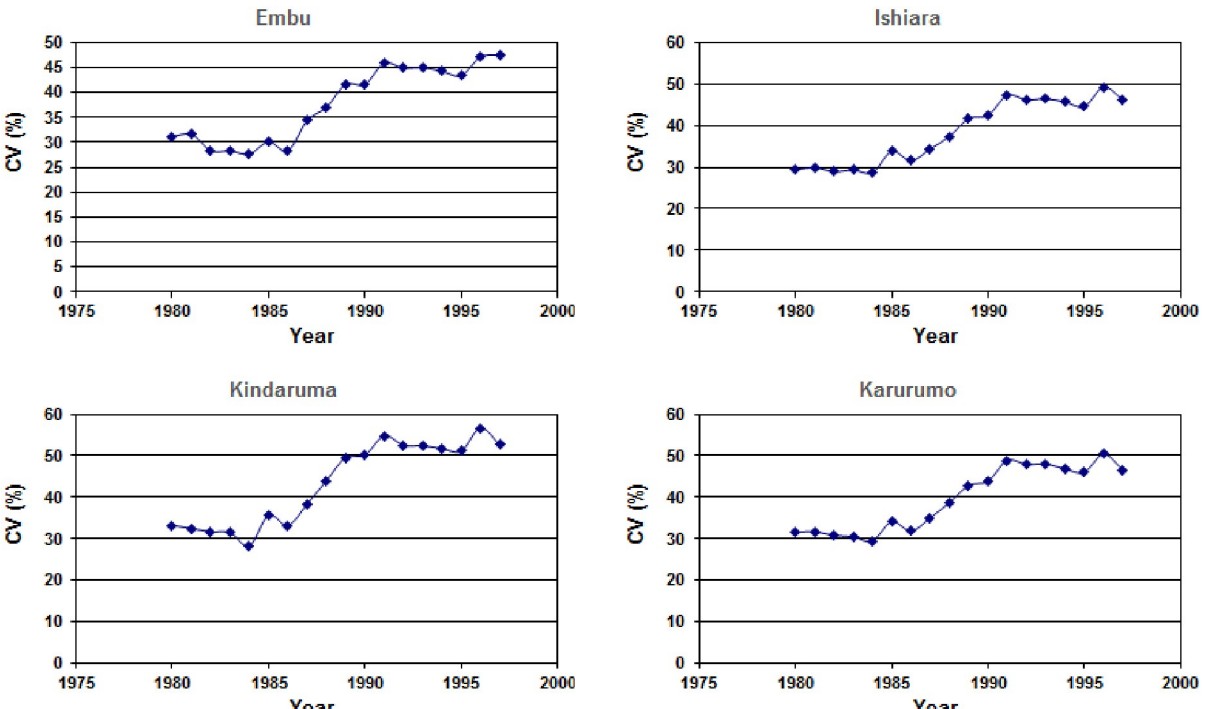

**Fig 4. Ten year moving coefficient of Variation (CV) of rainfall from 1980 during the short rain (SR) season at the four sites in Embu County, Kenya.** (The analysis is limited to 1997 since the data for remaining years is gap filled with AgMERRA data).

this change is substantially low under RCP 4.5 with Tmax varying between 0.50 and 2.70°C and Tmin between 0.70 and 2.90°C. On an average, projections under RCP 8.5 are higher by 1.68°C for Tmax and 2.04°C for Tmin compared to the projections under RCP 4.5 to end century. Most models projected greater increase in Tmin than Tmax. Only four of the 20 GCMs— BNU-ESM, GFDL-ESM2G, GFDL-ESM2M and IPSL-CM5A-LR predicted higher increase in Tmax than Tmin under RCP 8.5 emission scenario. Eight GCMs predicted more than 4.00°C increase in Tmax to end century under RCP 8.5 while 13 GCMs predicted more than 4.00°C increase in Tmin. The projected increase in Tmin to end century under RCP 8.5 is 4.0°C which is 0.50°C higher compared to the projected 3.50°C increase in Tmax. The projected climate shows a clear shift from the current temperatures which leads decrease in the frequency of low temperature events (below 9.00°C for Tmin) and increase in the frequency of high temperature events (above 34.00°C for Tmax). This shift may push the temperature range outside the optimum range for some of the crops grown in the study area (Fig 6).

Compared to temperature, changes in projected rainfall are more uncertain. Annual rainfall projections by different GCMs to end-century varied from -25% to 111% under RCP 8.5 and from -18 to 71% under RCP 4.5 to end century (Fig 5). However, majority of the GCMs, 15 of the 20, predicted an increase in rainfall amounts across the four study locations. Among the GCMs, GFDL-ESM2G, MIROC5, and NorESM1-M project 8 to 25% decline in rainfall while CanESM2, IPSL-CM5A-MR and BNU-ESM project more than 85% increase in rainfall by end-of-century under RCP 8.5. On an average, rainfall is expected to increase by 32.5% under RCP 8.5 and by 16.7% under RCP 4.5 to end century.

The projected changes in rainfall have further indicated a greater increase in rainfall during SR season compared to LR season under RCP 8.5 to end century period. Rainfall during the SR season varied from -16 to 241% with an average of 96% while that during LR season varied

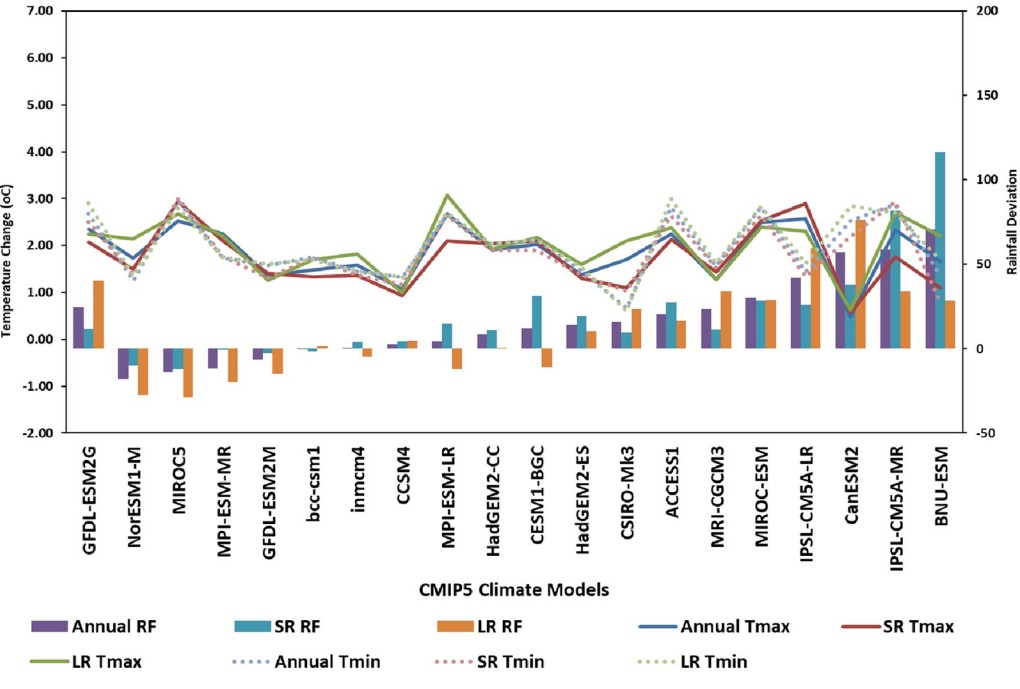

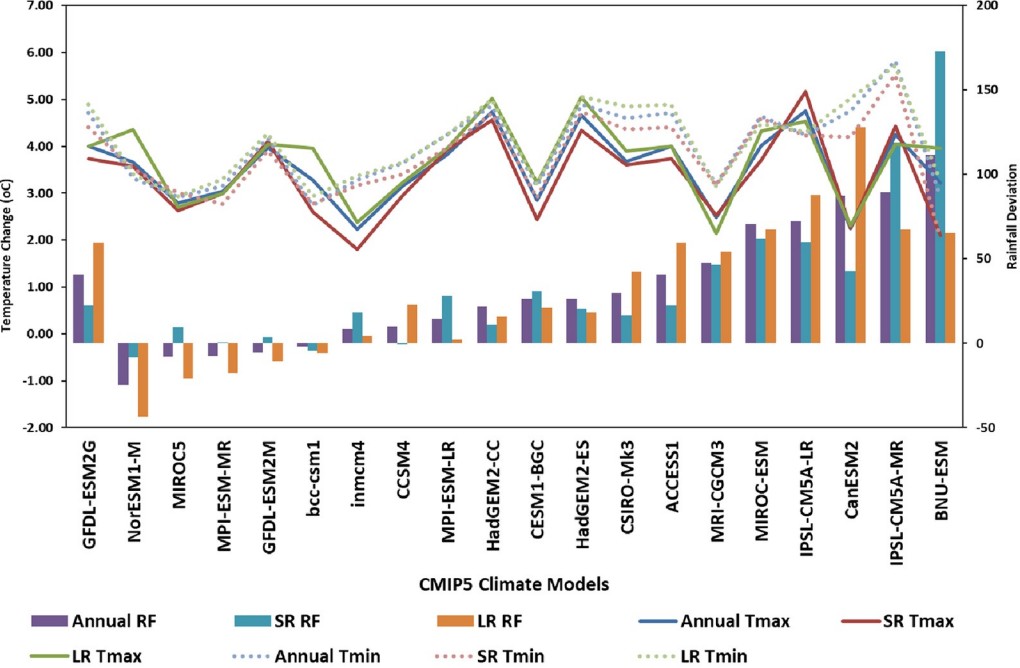

**Fig 5. Projected changes in maximum and minimum temperatures (absolute change) and in rainfall (percent deviation from historie rainfall) by 20 GCMs under RCP 4.5 (upper) and 8.5 (lower) by End- century for Embu, Kenya.**

from -9% to 171% with an average of 32% to end century under RCP 8.5. In case of RCP 4.5, the changes are relatively small compared to those with RCP 8.5 and no major difference was observed in the projected rainfall amounts during SR and LR seasons. The amount of rainfall projected by different GCMs varied between -29 and 76% with a mean of 12% during SR

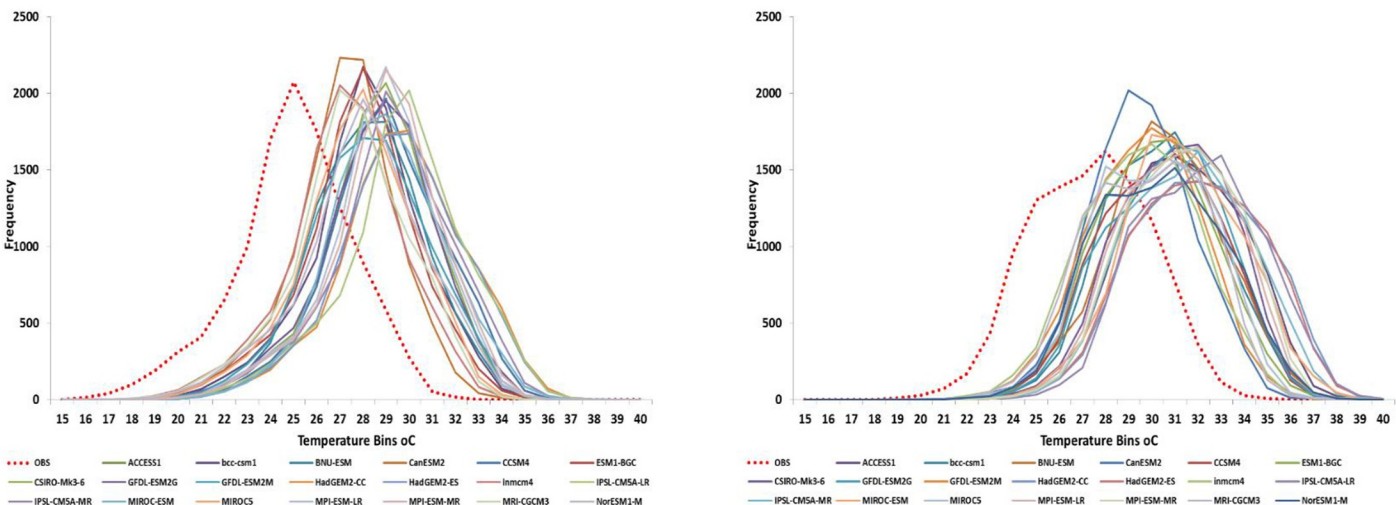

**Fig 6. Probability density function of projections in maximum temperature by GCMs at Embu (left) and Ishiara (right) in Embu county, Kenya (red dotted line denotes observed temperature).**

season and between -10 and 116% with a mean of 20% during LR season. Almost all GCMs with the exception of GFDL-ESM2G predicted higher increase in SR season rainfall compared to LR season rainfall under RCP 8.5.

**3.2.1 Calibration and validation of APSIM and DSSAT.** The maize varieties H511, H513 and Katumani were selected to represent the cultivars used by farmers in the Embu county. APSIM and DSSAT crop models were calibrated for these maize varieties using the experimental data as discussed in the material and methods section. The observed crop phenology, biomass and yield are satisfactorily simulated by the models for three cultivars at Embu. The model's ability to reproduce the phenological and yield attributes were tested using experimental data for three maize cultivars. The rates of phenological development were calibrated well with 6–7% RMSE (Root Mean Square Error) while, at flowering and maturity RMSE is observed between 2–4% as displayed (Table 6). The simulated values for total aboveground biomass and grain yield were quite close to the observed values (Table 6).

The CSMs are validated using 160 farmers crop data which fall under Embu climate. Both the CSMs simulated maize yields are higher than observed yields in the region. The differences between simulated and observed yields varied from as little as 20 Kg/ha to as high as 4000 kg/

**Table 6. Observed (average of three seasons) and DSSAT and APSIM modeled phenology and grain and biomass yields of three maize varieties.**

| Variety | Flowering | | | Maturity | | | Biomass | | | Yield | | |
|---|---|---|---|---|---|---|---|---|---|---|---|---|
| | Observed | DSSAT | APSIM | Observed | DSSAT | APSIM | Observed | DSSAT | APSIM | Observed | DSSAT | APSIM |
| H511* | 68.7 | 71.3 | 71.3 | 137.7 | 139.7 | 142.0 | 12495.7 | 12082.0 | 11580.0 | 4677.3 | 4428.7 | 4654.3 |
| H513* | 73.3 | 71.3 | 72.3 | 141.0 | 137.0 | 137.0 | 13391.3 | 13681.7 | 13479.3 | 5282.7 | 5027.7 | 4597.3 |
| Katumani* | 53.0 | 54.0 | 50.0 | 103.5 | 105.5 | 104.0 | 8567.5 | 8088.5 | 8888.0 | 4060.5 | 4058.5 | 3911.0 |
| Mean | 66.6 | 67.8 | 67.2 | 134.8 | 134.6 | 134 | 12884.0 | 12454.6 | 12567 | 5037.2 | 4890.6 | 4794.8 |
| SD | 9.7 | 9.0 | 10.2 | 16.8 | 15.8 | 17.8 | 1938.5 | 2391.0 | 2465.9 | 1097.7 | 1112.0 | 1035.6 |
| CV | 14.6 | 13.3 | 15.2 | 12.5 | 11.7 | 13.3 | 15.0 | 19.2 | 19.6 | 21.8 | 22.7 | 21.6 |
| RMSE | | 2.76 | 2.49 | | 3.19 | 3.52 | | 783.8 | 747.1 | | 318.7 | 357.2 |

SD = Standard Deviation, CV = Coefficient of variation and RMSE = Root mean square error.

ha. This could be attributed to various factors such as differences in interpreting and translating farmer description of the resource endowment into model parameters, inability of the models to capture the effects of biotic stresses such as pests, diseases and weeds, inaccuracies in estimating per hectare yields from bags per plot as reported by farmers and inaccuracies in defining the initial conditions. However, the simulated long-term yields of different AEZs reflected the trends in the yields reported by farmers fairly well, especially in the low potential LM4 and LM5 AEZs. In these AEZs, high moisture stress is the major yield limiting factor and this to a large extent masks the relatively low effect of other management practices and also the influence of differences in the resource base.

Using parametrized APSIM and DSSAT crop simulation models, impact of climate change on maize production in the five AEZs is assessed. Maize yields varied in response to differences in the magnitude of projected change in surface temperatures and rainfall by different GCMs under different scenarios. Maize yields also varied depending on the AEZ, season, and management practices. While there is a general agreement in the trends in maize yields simulated with APSIM and DSSAT under different climate change scenarios and AEZs, there are differences in the magnitude of the impact on maize.

**3.2.2 Impacts across AEZs.** CSMs simulations maize yields indicate that the impacts of climate change on maize yields are largely positive in the high potential AEZs of UM2, UM3 and LM3 and negative in the low potential AEZs of LM4 and LM5 (Fig 7). However, the magnitude of this change is higher in the yields simulated with DSSAT compared to that with APSIM. For example, the increase in DSSAT modelled maize yields to end century under RCP 8.5 varied between 3.6 and 47.6% in the high potential LM3 and between -43.0 and -17.5% in low potential LM4 over corresponding baseline yields. For the same scenario, APSIM simulated yields varied between 6.3 and 15.0% in case of LM3 and between -21.3 and -7.8% in LM4. In general, the impacts of climate change on maize yields modelled with DSSAT are more positive for the projections by GCMs CanESM2, BNU-ESM, IPSL-CM5A-LR, MIROC-ESM, and MRI-CGCM3 while negative with projections by GFDL-ESM2G, INMCM4, MPI-ESM-MR, BCC-CSM1, GFDL-ESM2M and MIROC5. In case of APSIM, projections by CESM1-BGC, MRI-CGCM3, GFDL-ESM2M, NorESM1-M and MIROC5 showed greater positive impact compared to IPSL-CM5A-LR, IPSL-CM5A-MR, CanESM2, BNU-ESM and MIROC-ESM.

**3.2.3 Impacts across seasons.** Impacts of climate change on maize yields were also found to be different in the two crop growing seasons. The impacts were more positive during SR season compared to LR seasons. Average maize yields during SR season to end century under RCP 8.5 increased by 14.5% with DSSAT and by 1.5% with APSIM and declined by about 1.0% during LR season with both the models. In both seasons, maize yields followed the general trend of predominantly positive changes in the high potential AEZs and negative changes in the low potential AEZs. Average annual (both seasons) maize yield modelled with DSSAT by end century under RCP 8.5 varied between 13 and 48% (with an average deviation of 30%) in LM3 and between -46 and -10% (with an average deviation of -31%) in LM4 compared to baseline yields. In case of APSIM, the increase in maize yields in LM3 varied between 6 and 18% (with an average deviation of 12.7) and the decline in LM4 ranged between -20 to -12% (with an average deviation -15.4%). The magnitude of either positive or negative change in maize yields under different AEZs is in the order 4.5 MID, 4.5 END, 8.5 MID and 8.5 END which is also the order in which changes in temperature and rainfall have occurred.

**3.2.4 Impacts across maize cultivars.** Among the maize varieties, Katumani, a short duration local cultivar that is widely adopted by farmers in Kenya, was found to be more sensitive to the changes in the climate compared to the other three varieties. APSIM simulated yields of Katumani under RCP 8.5 to end century declined by 5% during SR season and by 9% during LR season. Yields of other three varieties increased by 3–7% during SR season and

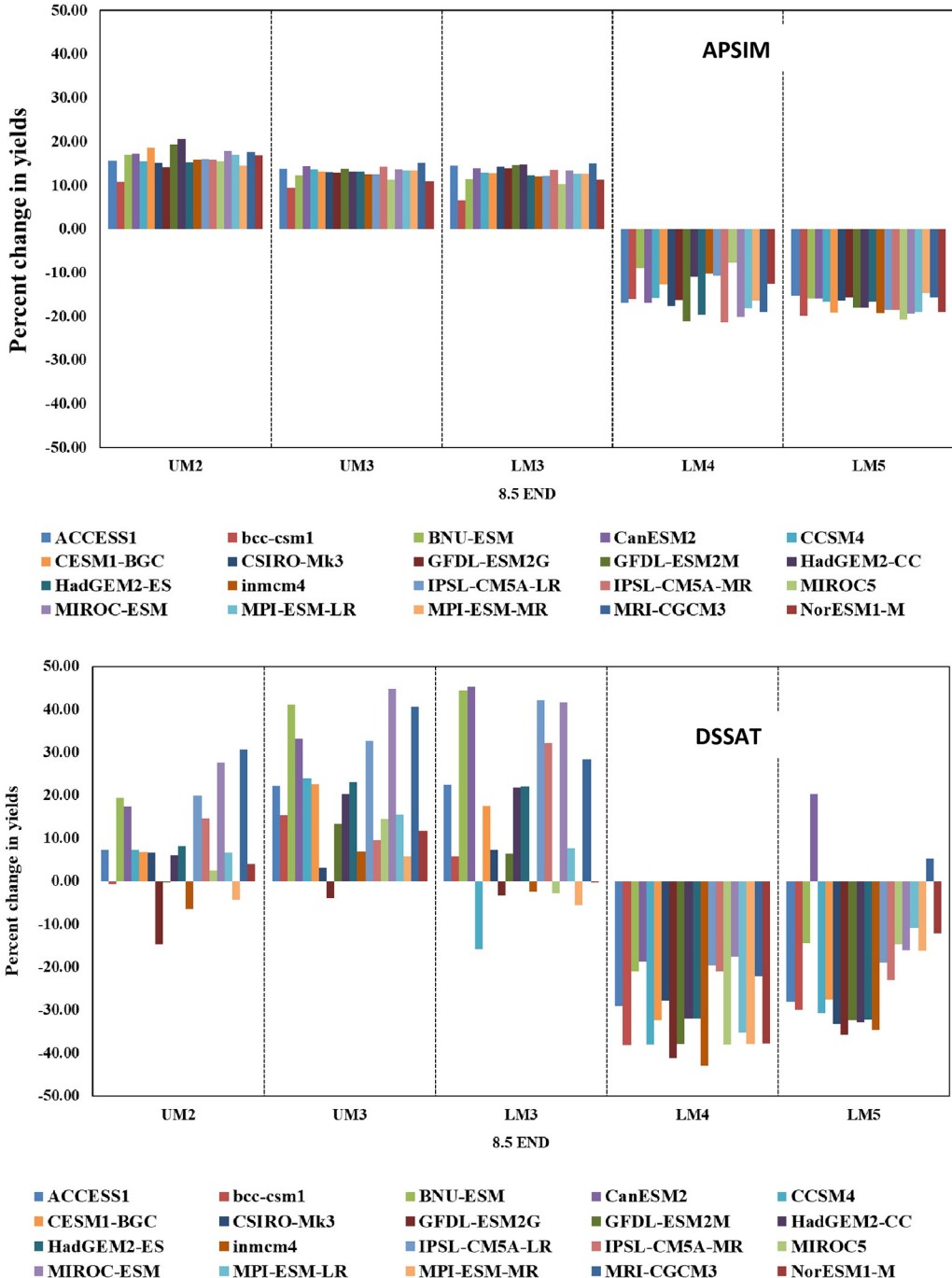

**Fig 7. Changes in maize yields (%) from baseline as simulated by APSIM (above) and DSSAT (below) with future climàtic conditions from 20 GCMs by end Century under RCP 8.5 in different agro ecological zones of Embu county, Kenya without elevated C02.**

varied between -1.5 to 1.4% during LR season. When modelled with DSSAT for the same scenario, yields were increased by 13% during SR season and by 1.7% during LR season. Yields of other varieties increased by 21–38% during SR season and by -4.5 to 2.4% during LR season (Fig 8). The results further indicate that the duration of crop is getting shortened by 6–15 days

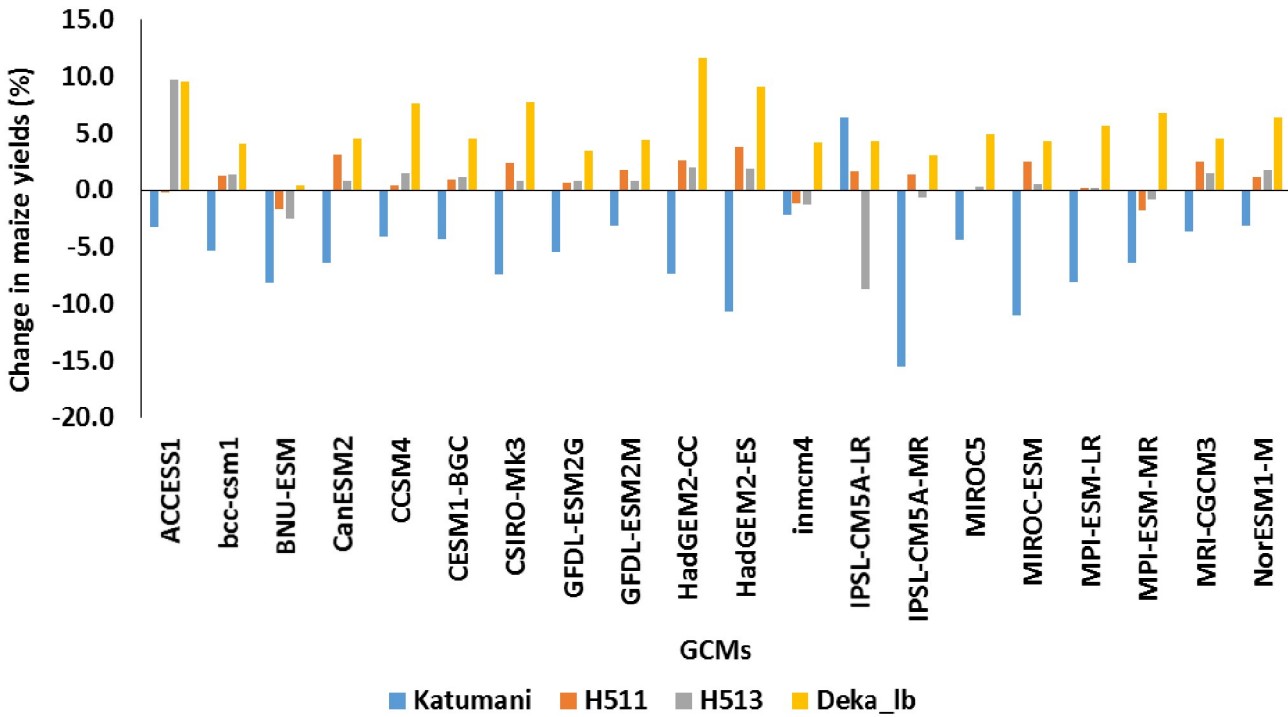

**Fig 8. Impact of climate change on the performance of different maize varieties under cultivation in different AEZs in Embu county of Kenya.**

for each degree increase in the average temperatures. This reduction in the duration of the crop has bigger effect on short duration Katumani cultivar compared to long-duration varieties H513, H511 and DEKA_LB which were able to maintain the same yield levels. Hence, replacing short-duration varieties with long-duration varieties is considered one of the options for adapting to climate change.

**3.2.5 Impacts of planting time.** APSIM and DSSAT simulated yields were found to be higher when planted between mid-March (day of year 74) and mid-April (day of year 105) for LR season and between start of October (day of year 274) and start of November (day of year 305) for SR season under baseline climate. Under climate change, mid-March (day of year 74) to end March (day of year 89) for LR season and start of November (day of year 305) to mid of November (day of year 319) for SR season was found to be optimal time for planting maize. The results also indicate that maize planted early within the identified planting window performed better during both SR and LR seasons. Early planted maize (Table 7) yields were found to be 25% higher in SR season and 7% in LR season to end century under RCP 8.5 with

**Table 7. Adaptation strategy for different agro-ecological zones with best combination of planting time, plant population, variety and fertilizer nitrogen for LR and SR seasons.**

| AEZ | Adaptation strategy for LR season | | | | Adaptation strategy for SR season | | | |
|-----|---------------|------------|---------|----------------------------|---------------|---------|------------|------------|
| | Planting Time | Plant pop. | Variety | Fertilizer nitrogen (kg/ha) | Planting Time | Variety | Plant Pop. | Fertilizer |
| UM2 | 15–30 Mar | 50 | H513 | 80 | 1–15 Nov | H511 | 40 | 70 |
| UM3 | 15–30 Mar | 50 | H513 | 70 | 1–15 Oct | H513 | 40 | 60 |
| LM3 | 15–30 Mar | 50 | H513 | 60 | 1–15 Oct | Deka_lb | 50 | 80 |
| LM4 | 15–30 Mar | 50 | Deka_lb | 60 | 15–30 Oct | H511 | 50 | 70 |
| LM5 | 15–30 Mar | 50 | H511 | 60 | 1–15 Nov | Deka_lb | 40 | 60 |

DSSAT. In case of simulations by APSIM, a smaller increase of 1.1% was recorded during SR season while declined by 2.3% during LR season with early planting.

**3.2.6 Impacts of plant population.** Among the other agronomic practices that significantly influenced maize yields is plant population. Both models simulated higher maize yields with increasing plant population. In SR season, maize yields increased from 1239 to 1481 kg/ha with DSSAT and from 1719 to 2718 kg/ha with APSIM when plant population is increased from 30,000 to 50,000 plants/ha. With 52% of the farmers surveyed using 30,000 plants/ha and other 42% using 40,000 plants/ha, most farmers are using below optimal plant population. Only 6% of farmers adopted 50,000 plants/ha which is also the recommendation for these areas. Furthermore, DSSAT simulations indicate an increase in maize yields in both SR and LR seasons with different plant populations. Maize yields increased by up to 16% with 30,000 plants/ha and 42% with 50,000 plants/ha during SR season and by 0.3 and -0.8% during LR season by end century under RCP 8.5. The increase in yield is higher with higher plant population in SR season and with lower plant population in LR season. APSIM simulations indicate less than 3% increase in maize yields when plant population increased from 30,000 to 50,000 plants/ha.

**3.2.7 Impacts of soil fertility and nitrogen application.** APSIM and DSSAT differed in simulating the impacts of climate change on maize yields on soils with different water and fertility regimes in SR and LR seasons. The kavuturi soil is the most fertile soil with an organic matter content of 2.29% followed by Gachuka (1.58%), Embu (1.49%) and Machanga (0.5%). Impacts of climate change are mostly positive on all soils during SR season and negative in LR season. Average increase in SR season maize yields under climate change on all soils is 3.6% with APSIM and 11% with DSSAT to end century under RCP 8.5. Similarly, maize yields during LR season increased by 4.2% with APSIM and by 9.9% with DSSAT. The magnitude of projected change in yields under climate change is higher with DSSAT relative to APSIM.

In all climate change scenarios, higher nitrogen levels increased maize yields. DSSAT simulations indicate 16–20% increase in SR season maize yields with fertilizer application up to 80 kg N/ha. While, APSIM simulations indicate less than 2% increase. In case of LR season. DSSAT simulated less than 2% reduction in the yields of maize with application of nitrogen up to 80 kg/ha. APSIM simulated yields to end century under RCP 8.5 increased by 2.6% with 0–20 kg N/ha and declined by 2% with 20-40kg N/ha and by 7% with 40–80 kg N/ha compared to the corresponding baseline yields.

## 3.3 Adaptation strategies

The differential impacts of climate change on maize under different management strategies in the five AEZs were further examined and used to frame an adaptation strategy by making adjustments to the current management by avoiding practices that are negatively impacted and adopting those positively responded to the projected changes in climate. Given that the impacts of climate change are going to be largely positive, the focus of this adaptation strategy is more on capitalizing on the benefits offered by changing climatic conditions. Accordingly, adaptation strategies were developed for each AEZ by identifying a set of management practices that included best performing maize cultivar, planting time, plant population and fertilizer amount (Table 7). Performance of this strategy under climate change was assessed by repeating the simulation analysis with both APSIM and DSSAT using the downscaled climate change scenarios from the 20 CMIP5 GCMs for mid and end century periods under RCPs 4.5 and 8.5.

DSSAT simulated maize yields with adapted crop management practices under all climate change scenarios are considerably higher than the current yields in all AEZs (Fig 9). However,

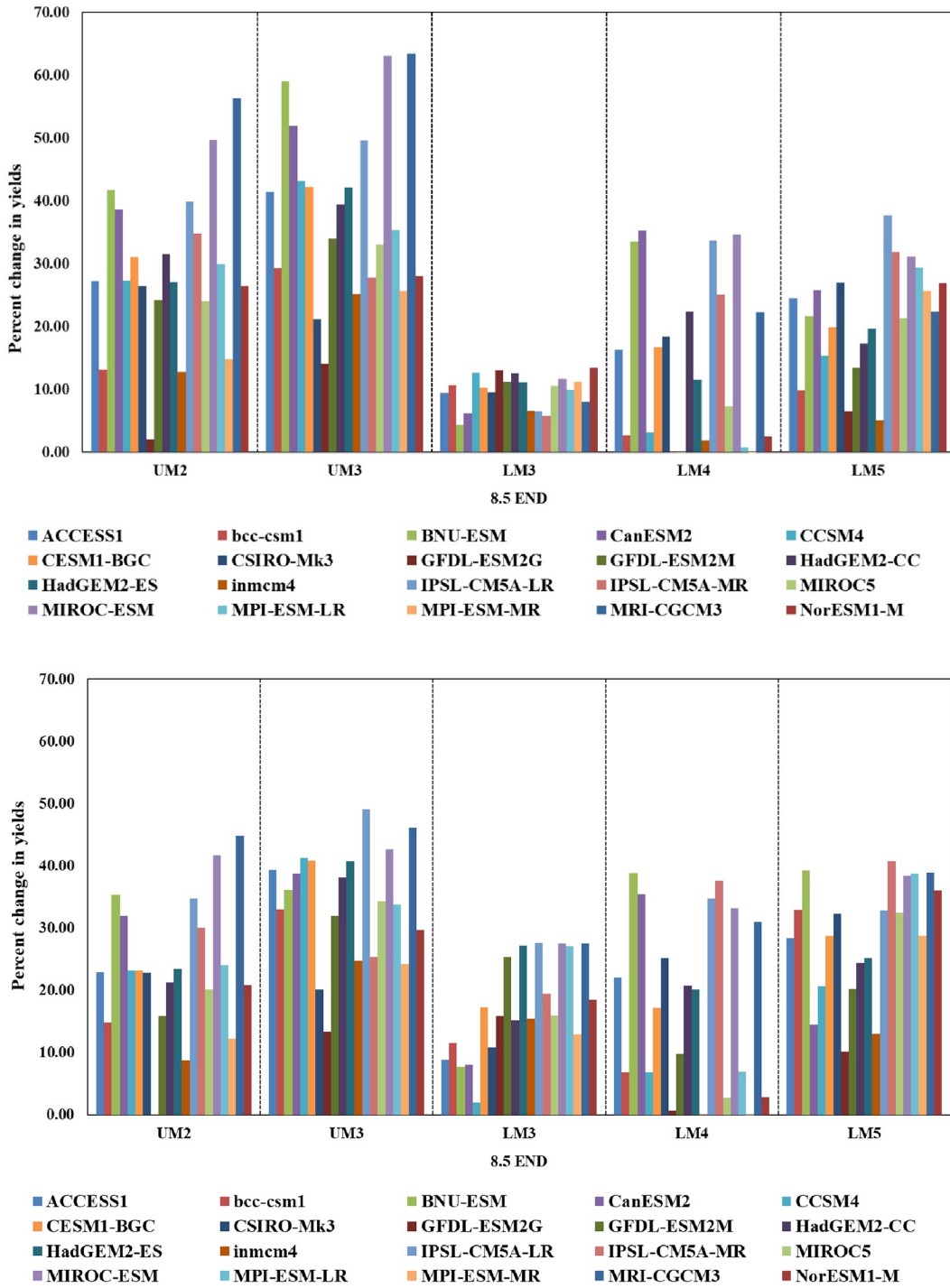

**Fig 9. Changes in maize yields** *(%)* **from baseline as simulated by APSIM (above) and DSSAT (below) with future climate projections from 20 GCMs by end Century under RCP 8.5 in different agro-ecological zones of Embu county, Kenya with elevated CO2.**

there are differences in the magnitude of the projected increase with GCM and RCP. In all the AEZs, the magnitude of this increase is higher under RCP 8.5 compared to 4.5 and by end century compared to mid-century. For example, maize yields in LM3 under RCP 4.5 increased by 140.7% during mid-century and by 156% during end century, while under RCP 8.5 yields

increased by 241.1% during mid-century and by 256% to end century. Among the AEZs, highest increase is observed in LM3 in which maize yield increased by more than 250% while the lowest increase is in UM2 in which yields went up by about 50% over the current levels with climate change by end century under RCP 8.5. Among the RCPs, yields simulated with projections under RCP 8.5 by GCMs MRI-CGCM3, MIROC-ESM, BNU-ESM and IPSL-CM5A-LR tend to fall in the upper quartile (75% percentile) and those simulated by NorESM1-M, MPI-ESM-MR, INMCM4 and GFDL-ESM2G tend to fall in the lower quartile (25 percentile).

Simulations with APSIM suggest that highest increase in maize yields will be in LM5 and lowest in LM3. According to DSSAT simulations, highest increase is in LM5 and least in LM3. Under climate change with the proposed adaptation strategies yields are expected to increase by about 128.2% in LM5, 121.1% in LM4, 64.7% in UM2, 85.2% LM3 and 61.1% in UM3 by end century under RCP 8.5 compared to the yields under current climate with current management as depicted in Fig 10. In case of APSIM, the difference in maize yields under climate change with adaptation to mid and end-century periods in all AEZs except LM3 is less than 54%. No clear trend in the response of maize to differences in the climatic conditions predicted by different GCMs is noted. The GCMs in the lower and upper percentiles in relation to change in maize yields are different for different AEZs and also for different emission scenarios. Maize yields with projections by NoRESM1-M for UM3 and LM3 and projections by CCSM4 for UM2, LM4 and LM5 are in the upper percentile while those with projections by BNU-ESM-4-5 and CANESM2 for UM3 and LM3 and by GFDL-ESM2G for UM2, LM4 and LM5 are in the lower percentile.

## 4. Discussion

### 4.1 Climate variability

Understanding the trends in historical observed climate is important for two reasons. Firstly, they help in understanding the sensitivity and robustness of the target systems to climate variability based on which impacts of projected changes in climate can be more realistically assessed. Secondly, they serve as a basis to evaluate the future projections generated by climate models (GCMs) which is an essential first step to assess the impacts of climate change on target systems. The results from the trends analysis have clearly highlighted the high variability in the rainfall and a clear increasing trend in Tmax and Tmin, particularly, during the last two decades across the study locations.

While no clear declining or increasing trend either in annual or seasonal rainfall was observed, evidence indicates increasing variability in rainfall during the past two decades. Increase in the variability of rainfall and more frequent occurrence of extreme events was also reported by some recent studies which are based on observed long-term rainfall data [45–47]. Evidence suggests that rainfall during SR season is relatively low and more variable compared to that during LR season. The increase in ten-year moving CV of SR rainfall from 30% in 1980s to 50% during 2000–10 is a major change with a potential to impact the productive potential of many crops. Assorted studies have cited similar patterns in seasonal rainfall over lower eastern Kenya [48–51]. The study has also established that surface temperatures are warming at the rate of up to 0.03˚C/decade. Increasing variability in rainfall together with warmer temperatures will have strong impact on seed germination, length of growing season, flowering and grain filling of most crops grown in lower eastern Kenya.

### 4.2 Climate change scenarios

Given the large uncertainty in the GCM projections from unknown future trajectories of $CO_2$ and $CH_4$ emissions and highly simplified representation of reality encoded in these models

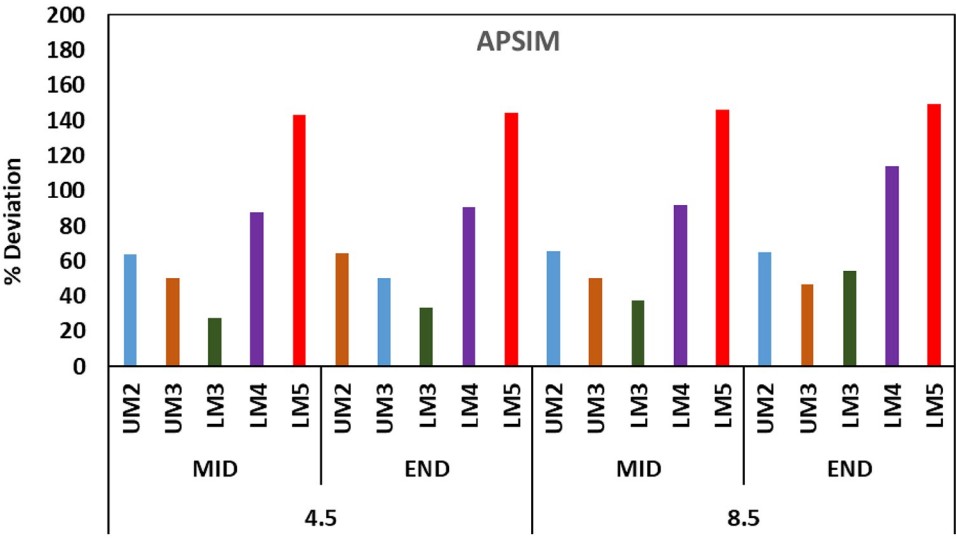

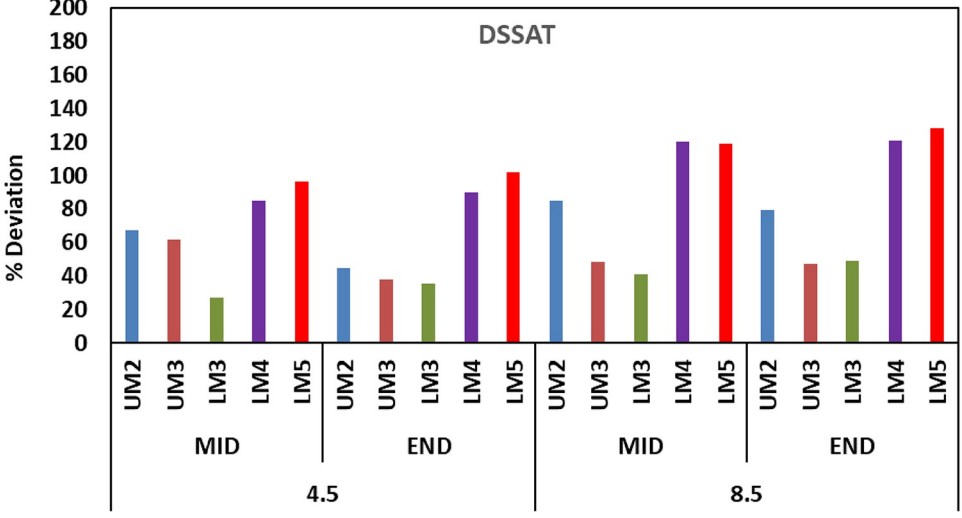

**Fig 10. Projected increase in maize productivity with adaptation compared to non-adoption in different agro-ecological zones under different climate change scenarios based on APSIM (above) and DSSAT (below) simulated yields.** The deviation is the percent increase in current yields com pared to average yield with projections by 20 GCMs.

[39], use of multiple models was suggested to provide more reliable assessment of impacts of climate change on weather sensitive sectors like agriculture [26,37,38]. In this study, we used outputs from 20 GCMs under two emission scenarios (RCP 4.5 and 8.5).

CMIP5 downscaled future climate projections have indicated significant increase in surface temperatures under the RCPs 4.5 and 8.5 to the end of 21[st] century. These projections are based on the expected changes in atmospheric $CO_2$ concentration which will be 499 ppm by mid-century under RCP 4.5 and 801 ppm by end century under RCP 8.5. The projected increase in temperatures is generally in the order of 4.5-mid<4.5-end<8.5-mid<8.5-end. The magnitude of increase in Tmin and Tmax projected by majority of the models to end century

under RCP 4.5 is equivalent to the predictions to mid-century under 8.5. All GCMs predicted a substantial warming by end century under RCP 8.5 which is almost double to what the models predicted to end century under RCP 4.5 or to mid-century under RCP 8.5. A sharp increase in the temperature under RCP 8.5 by end century was also reported in AR5 [9] which is based on the comprehensive review and synthesis of all available information. The report concludes that the multimodal median increase in Tmax over global land by end of the century will be 2.7˚C under RCP 4.5 and 5.4˚C under RCP 8.5. Surface temperatures over Africa are expected to rise faster than the rest of the world, by about 2.00˚C during mid-century and by 4.00˚C to end of 21st century [52]. However, there are differences in the magnitude of these increases between Tmax and Tmin and between annual and seasonal averages.

Majority of the GCMs project higher increase in Tmin than in Tmax. While Tmax is projected to increase by about 4.80˚C under RCP 8.5 to end of 21st century, the projected increase in Tmin under similar conditions is 5.80˚C. Further, the warming is more during SR season compared to that during LR season. Asymmetric changes in Tmin and Tmax and associated deviations in Diurnal Temperature Range (DTR) have been reported over the past three to five decades mainly because of the relatively stronger increases in daily Tmin than daily Tmax. This result closely corroborates the earlier findings which reported that the temperature is likely to increase by 3.00–5.00˚C in the African tropics during 2071–2100 relative to 1961–1990 under the high emission scenario [50–57].

Compared to temperature, it is more difficult to predict changes in rainfall due to high natural variability associated with it. It is for this reason AR5 assigns medium to low confidence to the past trends and future projections of rainfall. Most GCMs projected an increase in rainfall at all locations in both SR and LR seasons which is consistent with the projections for East Africa and to the locations close to equator [9,52,58]. IPCC AR5 report suggests the future precipitation projections are likely to increase in the eastern Africa and decrease in the southern part [52]. Various other studies have projected that rainfall over east Africa will increase [46,47,59]. Our results followed the reported trends for the region, with increase in rainfall in all zones. The increase is higher under RCP 8.5 than that under RCP 4.5. The projected changes in rainfall varied from -25% to 111% under RCP 8.5 by end of 21st century with greater increase during SR season compared to LR season. Although the projected changes in rainfall seem to favour the agro-ecological sector, associated variability in rainfall and warming also play an important role in determining the overall impact of changed conditions on crop production. For instance, despite projecting an increase in rainfall, [60] indicated that reduction in soil moisture content as a result of increased temperatures have contributed to a reduction in crop yields. An increase in rainfall without corresponding increase in number of rainy days leads to an increase in extreme rainfall events that can contribute to environmental degradation through increased runoff and erosion.

## 4.3 Impacts of climate variability and change on maize production

Our assessment of impacts of climate change on maize production using process-based models has provided useful insights into the climate sensitivity of the crop and how the projected changes in climate impact the productivity of the system along with associated uncertainties. These insights are extremely useful not only in understanding the impacts of climate change but also in developing locally relevant adaptation strategies. Under considerable uncertainty relating to future climate change and its consequences [61], this study offered a unique opportunity to assess impacts of climate change on highly diverse small holder agriculture under five AEZs. Significant variability is observed in the current maize yields in all the five AEZs and across all the farms mainly due to variation in the soil, cultivar and crop management

practices. Much of the past work on assessing the impacts of climate change on agriculture was carried out at national and continental level using statistical and empirical models that fail to account for the full range of complex interactions between various factors that contribute to the production and productivity of the agricultural systems and climate [13,62,63]. The key messages that emerged out of these large-scale assessments are about 65% of the current maize growing areas in Africa will experience yield losses [11] and the predicted production losses for most crops are in the range of 10–25% by 2050 [64]. These assessments are extremely useful in understanding the overall impacts of climate change on food security and suitable for developing adaptation strategies to overcome the projected losses. However, they will not be able to provide detailed information on how, where and when these impacts will occur and which environments gain or lose.

This assessment highlighted the differential impacts of climate change on maize yields across the growing environments and management practices which can serve as a useful basis in developing appropriate and well-targeted adaptation strategies. Among the growing environments, our study suggests that maize yields are going to increase in the high potential environments represented by AEZs of UM2, UM3 and LM3 and decline in low potential environments represented by AEZs of LM4 and LM5. The increase in maize yields in the high potential AEZs is mainly due to the temperatures remaining within the optimal range for maize production even with an increase of 2.50 to 4.80˚C. In UM2, UM3 and LM3 the current Tmax are around 24–25˚C which with climate change is moving closer to the optimal temperature of 30˚C [65]. Similar results were also noticed in a study by [66] in Ethiopia using APSIM and CERES maize models under 20 GCMs and RCP 4.5 and 8.5 reported an increase in maize yields between 1.7% and 4.2%. Based on the analysis of maize yields from a data set of more than 20,000 historical maize trials in Africa along with daily weather data, [11] concluded that each degree day spent above 30.00˚C reduced the final yield by 1% under optimal rain-fed conditions, and by 1.7% under stressed conditions. In addition to changes in temperatures, the GCMs on an average are projecting a 25–50% increase in rainfall by end century which is also a major contributor to the observed increase in maize yields under climate change.

The analysis further suggests that the impacts are more positive during SR season compared to LR season which is attributed to the higher increase and longer duration of rainfall. Under the current climatic conditions, SR season receives 20–30% higher rainfall than that during LR season and is more dependable with lower CV. Under climate change this difference will increase further since most GCMs project greater increase in rainfall during SR season compared to LR season. The average increase in rainfall projected by 20 GCMs under RCP 8.5 by the end century is close to 190 mm during SR season which is approximately double to the increase projected for LR season (105 mm). The duration of the rainfall is another important variable contributing to the seasonal differences in maize yield. In the target county Embu, the duration of SR season is longer with rainfall distributed over 120 days compared to LR season which receives rains over a 60-day period (Fig 11).

The analysis further highlighted the potential role soil and crop management practices can play in moderating the impacts of climate change. Among the management practices, impacts of cultivar, soil fertility and plant population were found to be quite important. Short duration maize cultivar such as katumani were found to be more negatively impacted compared to the longer duration maize cultivars H511 and H513. Temperature is a major determinant of the rate of plant development and, under climate change, warmer temperatures shorten the development stages, leads to earlier maturing of crops, which reduces time to accumulate biomass and form economic yield [67–70]. This results in a shorter life cycle leading to smaller plants, shorter reproductive duration, and lower yield potential [71]. According to the results from the simulation analysis the duration of the crop is expected to decline by about six days with

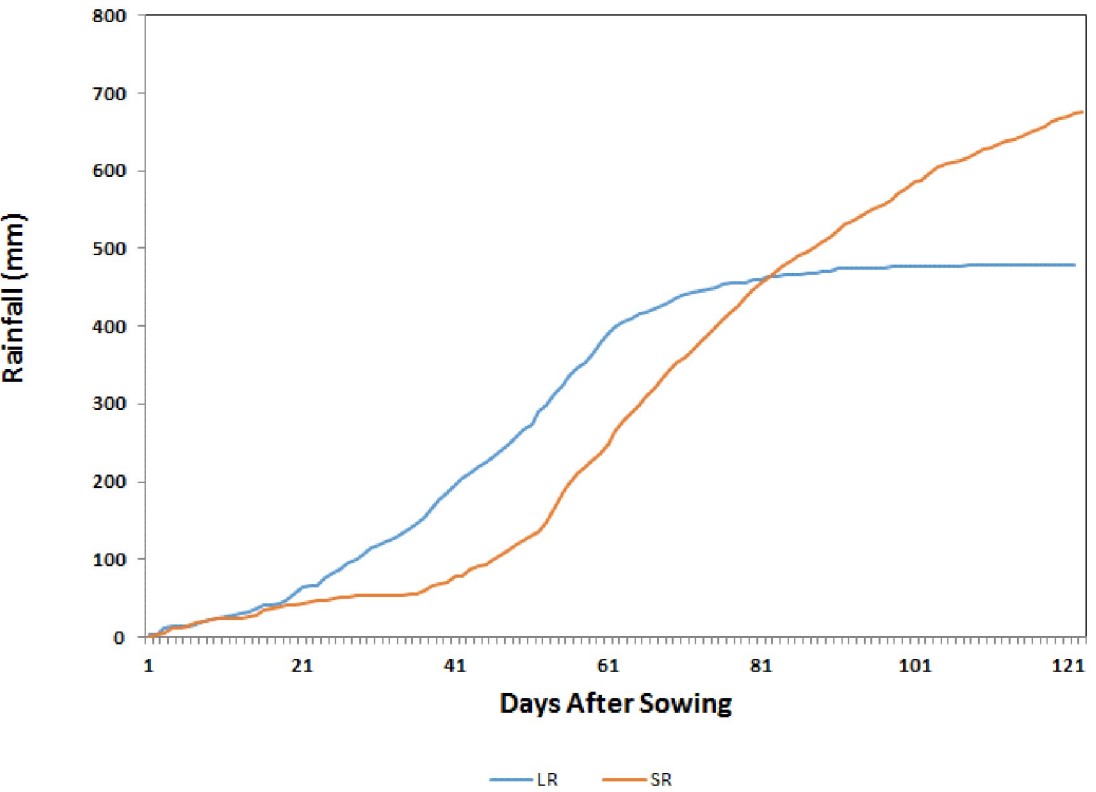

**Fig 11. Average cumulative rainfall during SR (Oct-Dec period) and LR (Mar-May) seasons at Embu, Kenya.**

every one-degree increase in Tmax between 25.00 and 30.00˚C which in case of short duration varieties is leading to a reduction in their yield potential. Under climate change, yields are increasing with increasing plant population in all AEZs. This can be attributed to either increasing environmental potential or reduced plant potential [72] both have a positive association with the required plant population. The other management factor that contributed significantly to the productivity of maize is the soil fertility. Highest increase is observed in fertile soils with high organic carbon content under low doses of fertilizer application. Similar increase in sorghum yields under low input systems was also reported by [73]. This was attributed to the greater availability of nitrogen with increased mineralization under warmer and wetter future climatic conditions.

## 5. Conclusions

In this paper we presented a robust approach to assess the impacts of climate change on maize production under a range of agro-ecological and management conditions. Despite the limitations in climate and crop models to simulate the systems accurately, this assessment has demonstrated that it is possible to make credible assessment of impacts of climate variability and change on smallholder farming systems that can aid in planning for adaptation. It helped in highlighting the impacts of variability in the current climate and projected changes to mid and end century periods on maize crop performance which formed the basis for developing strategies to adopt to the current and projected changes in climate. One important aspect of this study is in highlighting the differential impacts of climate change which contradicts the general perception that climate change always leads to negative consequences. The impacts of climate

change are different in different AEZs, seasons and management conditions. It also stresses the need for changing current management practices which are inadequate to capitalize on the changes in climate which in most AEZs are turning to be more favourable for maize cultivation. Significant productivity gains are possible by adopting available technologies and recommended management practices even under current climatic conditions. We used the household survey data and noticed that more than 50% of the farmers are using a plant population of 30,000 plants/ha and less than 25 Kg N/ha, much below the recommended population of 50,000/ha and 40 kg N/ha. Hence, the gain in maize yield with the use of adaptation strategy, which includes higher plant population, increased amount of nitrogen fertilizer and long-duration maize cultivar is partly due to the improved management and partly due to the changes in the environment which turned out to be more favourable for maize production with climate change than under current climate. There is a need for policy makers and practitioners to understand the differential impacts of climate change on maize cropping system in Embu county and prioritize the interventions aimed at adapting to climate change in a way that helps on capitalizing the positive changes and minimizing the negative impacts. Our study emphasizes the need for careful assessment of impacts of climate change with due consideration to the diversity in smallholder farm resources and developed adaptation strategies that are tailored to the local specific needs. Though the current assessment is limited to maize, the same can be extended to other crop enterprises and cropping system. Substantial progress is needed in CSMs to make them more robust for simulating the influence of weeds, pest and diseases on crop growth and development. Crop simulation models do not account for damage caused by pests and diseases on agricultural systems which is an important element in understanding the potential impacts of climate change on agricultural systems. Climate change is one factor driving the spread of pests and diseases and is expected to further increase surface temperatures along with rainfall favouring the growth and distribution of most pest and diseases by providing a warm and humid environment necessary for their growth and multiplication. Improving the capabilities of CSMs for estimating the pest population, damage level will enable researches to address possible impacts of biotic stress along with abiotic stress under changing climate. These efforts should go beyond the crop model community and include expertise on pests and diseases to enable capabilities of CSMs in assessing the impacts of climate change more accurately.

## Author Contributions

**Conceptualization:** Sridhar Gummadi, K. P. C. Rao.

**Data curation:** Richard Mulwa, Mary Kilavi.

**Formal analysis:** Sridhar Gummadi, M. D. M. Kadiyala.

**Investigation:** Sridhar Gummadi, Gizachew Legesse.

**Methodology:** Sridhar Gummadi, M. D. M. Kadiyala, K. P. C. Rao, Mary Kilavi, Gizachew Legesse.

**Project administration:** K. P. C. Rao.

**Resources:** M. D. M. Kadiyala, Ioannis Athanasiadis.

**Software:** Sridhar Gummadi.

**Supervision:** K. P. C. Rao.

**Validation:** K. P. C. Rao, Richard Mulwa.

**Visualization:** Gizachew Legesse.

**Writing – original draft:** Sridhar Gummadi.

**Writing – review & editing:** Sridhar Gummadi, M. D. M. Kadiyala, K. P. C. Rao, Ioannis Athanasiadis, Richard Mulwa, Mary Kilavi, Gizachew Legesse, Tilahun Amede.

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
