## [Decision Letter · Decision Letter 0]

24 Jun 2020

PONE-D-20-14901

Simulating possible Impacts of climate variability and change on Maize production in Embu County, Kenya

PLOS ONE

Dear Dr. Gummadi,

Thank you for submitting your manuscript to PLOS ONE. After careful consideration, we feel that it has merit but does not fully meet PLOS ONE’s publication criteria as it currently stands. Therefore, we invite you to submit a revised version of the manuscript that addresses the points raised during the review process.

We look forward to receiving your revised manuscript.

Kind regards,

Shamsuddin Shahid

Academic Editor

PLOS ONE

Journal Requirements:

'No'

'No'

6. Please amend either the title on the online submission form (via Edit Submission) or the title in the manuscript so that they are identical.

7. Please amend the manuscript submission data (via Edit Submission) to include author Tilahun Amede

8. Please amend your authorship list in your manuscript file to include author Ioannis Athanasiadis

Additional Editor Comments (if provided):

Please see the reviewers' comments. Both the reviewers find the article interesting. However, both think major revisions are required before publication. Particularly, first reviewer is very critical. Authors need to address all the comments of both the reviewers.

Reviewers' comments:

Reviewer's Responses to Questions

**Comments to the Author**

1. Is the manuscript technically sound, and do the data support the conclusions?

Reviewer #1: Partly

Reviewer #2: Yes

2. Has the statistical analysis been performed appropriately and rigorously? 

Reviewer #1: No

Reviewer #2: Yes

3. Have the authors made all data underlying the findings in their manuscript fully available?

Reviewer #1: Yes

Reviewer #2: No

4. Is the manuscript presented in an intelligible fashion and written in standard English?

Reviewer #1: Yes

Reviewer #2: Yes

5. Review Comments to the Author

Reviewer #1: General Comments

This is an interesting article assessing the impacts of climate change on growth and performance of maize using DSSAT and APSIM under RCP4.5/RCP8.5 at the mid and end century. However, the article can be improved by having a reduce and proper structure (section and sub-section) and headings. Some of the section/sub-section was found to be too general and not specific (can be combine with other sub-section) with the research title and objectives. Many sub-section and paragraph was found to be repetitive and redundant. It is preferable to have a sub-section with a heading directly discussing the objectives of the study.

I would strongly advice that a proper step by step procedure on the methodology involved should be prepare to clarify how the objectives was achieved. It was difficult to get a clear picture of the future projection (Tmin, Tmax and Prec) for the study because the authors are employed the 20 GCMs individually. The methodology can be improved by using an ensemble of GCMs, instead of using individual GCMs which introduce a lot of uncertainty in your final output. I would advise the authors to make an ensemble of the 20 GCMs or make an ensemble of the selected GCMs which can better modulate the local climate. Otherwise, the future projection output will have a big uncertainty, confusing and meaningless. The high uncertainty in the output, where some GCMs give positive results, while other GCMs give negative results, would render the results to meaningless as it seems unreliable to at least predict a certain degree of certainty in the future. It is understandable that different GCMs will have its own inherent variability. Therefore, an ensemble methodology should be use to reduce such uncertainty.

It is not clear how the authors get the results for certain output, either through individual GCMs or taking a mean based on 20 GCMs (it should be ensemble, not mean). Due to lack of clarity in the methodology involved in getting the final output of certain section/sub-section, the authors seem to be confuse about what it means by GCM ‘ensemble’.

It would be a major change in the results/output of the research if a proper ensemble method or any method selecting the best GCMs was applied, instead of using 20 individual GCMs (which introduce high uncertainties). I would advise for major revision of the article at its current state because without improve methodology, there will be a high uncertainty in the result/output of this article. Other specific comments can be found below.

Specific comments

Abstract:

• Line 9: Website link address can be insert at the Materials and Methods section

• Line 11: It then develops an adaptation strategy that is locally > Then, adaptation strategy was develop that is locally...

• Line 12: that help offset > to offset

• Line 15: but differed in the magnitude of that increase > with different magnitude

• Line 16: higher and those by CanESM2 > higher than CanESM2, INMCM4 and NorESM1-M under both emission scenarios.

• Line 18: abbreviations for minimum temperature (Tmin) and maximum temperature (Tmax) can be use here. Please change for the rest of the manuscript.

• Line 24-27: This sentences is too long to be in the abstract and confusing. What is the meaning of 'short duration varieties' and 'long duration varieties'? Do you mean ‘short duration maize varieties’…?

• Line 34: at national and local levels and make their promotion easier > at national and local level.

• Line 35-38: This sentence is too long. Need to be rephrase and simplify.

Introduction

• Line 57: plant available water? Is this a scientific term?

• Line 107: are viable is required > are viable.

• Line 122: to adapt to the same > for adaptation

• Line 124-126: Developing downscaled location specific climate change scenarios > Downscaling site specific climate change scenarios for various agro-ecological zones (AEZs) of Embu county in Kenya under RCP4.5 and RCP8.5 at the mid and end century period.

• Line 127: on the performance of maize > on maize yields

• Line 128: and to identify key vulnerabilities? What do you mean here? To identify vulnerable area or to identify vulnerability factor?

• Line 129: the objective is for a general agricultural systems or specific to maize agricultural system?

Materials and Methods

Study Location

• Line 157: potential area for what? agriculture? which crop?

• Line 161: do you mean long duration maize varieties?

Crop data and model parameterization

• Line 241: required data > the required data

• Line 248: and the same were derived from the data collected > and the same data was collected...

• Line 249: were derived from the amount > were based on the amount

• Line 265: NO3 and NH4 > NO3 and NH4

• Line 267: 50 mm and 0.75 m

Results

Observed trends in climate

• Line 278: also in the temperatures > also in the precipitation

Projected changes in future climate

• Line 326: for the crops grown in that AEZ > for the crops grown in the study area.

• Line 341: I think there is a major difference as well between SR and LR season under RCP4.5 although the difference under RCP8.5 is bigger.

Impacts of climate change on maize yields

• Line 389: to mid and end century > at the mid and end century

• Line 403-405: This sentence is redundant. Please remove.

• Line 406-412: It is better to have an ensemble of future projection to reduce the uncertainty rather than using and discussing each individual GCMs which could entail lengthy and contradictory results.

• Line 415-: How did you get the exact percentage result here? Previously, you discuss that there is a range of output by different 20 GCMs being used. You should not take the mean of the output from each GCMs here. Only an ensemble can do.

Effect of management on climate change impacts on maize yields

• Line 429: the effect of variety > the effect of maize variety

• Line 434-: The different and contradictory result might be due to the usage of 20 GCMs individually, instead of using an ensemble of GCMs. This might be the source of high uncertainty in your output by APSIM and DSSAT.

• Line 442: simulation analysis > simulation analysis,

• Line 477: What daoes it means by early planted maize? Can you specify the exact date and duration here?

Impacts under elevated carbon dioxide levels

• Line 493: It is better to have an ensemble of GCMs, rather than individual GCMs result as it will introduce high model uncertainty in your DSSAT and APSIM modelled yields.

Adaptation strategies

• Line 510: performing variety > performing maize variety

• Line 535: which propose intervention do you mean here?

Discussion

• The first paragraph is just a general statement and can be remove or relevantly insert at various part of other section/sub-section.

• Line 552: effectiveness > the effectiveness

Climate variability and change

• This sub-section is too general and doesn't reflect with the research title and doesn't reflect with the research objectives on maize yields. It is better to directly integrate the results being obtained here and relate it with other relevant sub-section which reflect your main findings (related to maize yield).

• The first paragraph is redundant. It is better to directly discuss your results.

• The second paragraph should be discuss in data section

• Line 594: Daily Temperature Range (DTR) > Diurnal Temperature Rang (DTR)

• Line 611: 'we used an ensemble of 20 GCMs' > This statement is confusing with your previous methodology and discussion. Are you using GCMs individually or an ensemble of GCMs? The author should have a proper understanding of what 'ensemble' means.

• Line 612-613: This statement should be discuss in the methods and data section. The reason for selecting only two different emission scenario (RCP4.5 and RCP8.5) and two time period (mid and end century) should be properly justify.

• Line 615-: This section should be in the methods section. It is repetitive.

Impacts of climate variability and change on maize production

• The statement in the first paragraph is general making it lengthy. It should directly discuss the sub-section topic on 'Impacts of climate variability and change on maize production'

Conclusions

• The first paragraph is redundant and repetitive from previous section. It should be remove.

• Line 745: This is the first time that 'multi-model ensemble' being mention, and it was mention at the conclusion. This is confusing. Your results discuss individual GCMs, not an ensemble of GCMs. Please clarify.

• Line 763-: The specific discussion and specific results for the intervention (changing variety, plant density, soil fertility management etc.) are scatter across the article and not properly and directly discuss in sub-section ‘Adaptation strategies’. It is difficult to understand how these interventions play a key role in maize production in relation with climate change impact.

Reviewer #2: The manuscript presents an approach to assess the climate change impact on Maize production, which is interesting. The subject addressed is within the scope of the journal; however, the manuscript, in its present form, contains several weaknesses. Appropriate revisions to the following points should be undertaken to justify the recommendation for publication. So, I suggest that this manuscript should be revised as major revision.

Abstract:

1- L9: Eliminate “(www.agmip.org)”.

2- There are mixed verb tenses, somewhere use the present tense “L8: The study assesses…..”, and other places use the past tense, “L12: The study used…”. Please use the present or past tense in the whole of the paper.

3- Please rewrite lines 13 to 15.

4- It would be better to highlight your contribution, more clearly in abstract.

Introduction:

5- The authors should highlight the research significant, more clearly in the introduction section.

6- There are some grammatical errors in this section. Please check whole the section carefully.

7- References are not always relevant and up-to-date. This reviewer suggests the inclusion of a few relevant publications. The list is given below:

https://rmets.onlinelibrary.wiley.com/doi/abs/10.1002/joc.6307

https://link.springer.com/article/10.1007/s11368-020-02632-0

https://link.springer.com/article/10.1007/s00704-019-02979-6

https://www.sciencedirect.com/science/article/pii/S0168192319302199

Method:

8- It is mentioned in L133 that the Embu County in Kenya is adopted as the case study. What are other feasible alternatives? What are the advantages of adopting this particular case study over others in this case? How will this affect the results? The authors should provide more details on this

9- It is mentioned in L181 that historical records of 1980 to 2010 are taken. Why are more recent data not included in the study? Is there any difficulty in obtaining more recent data? Are there any changes to the situation in recent years? What are its effects on the result?

Results and discussion

10- From Figure 6, it is evident that the authors obtained poor performance (R2: 0.33-0.44) during the validation phase. Is it a reliable model for crop simulating?

Conclusion

11- In the conclusion section, the limitations of this study suggested improvements in this work, and future directions should be highlighted.

6. PLOS authors have the option to publish the peer review history of their article (what does this mean?). If published, this will include your full peer review and any attached files.

Reviewer #1: Yes: Zulfaqar Sa'adi

Reviewer #2: No

---

## [Author Response · Author response to Decision Letter 0]

13 Aug 2020

Reviewer #1: (PONE-D-20-14901) Simulating possible Impacts of climate variability and change on Maize production in Embu County, Kenya.

General comments

 This is an interesting article assessing the impacts of climate change on growth and performance of maize using DSSAT and APSIM under RCP4.5/RCP8.5 at the mid and end century. However, the article can be improved by having a reduce and proper structure (section and sub-section) and headings. Some of the section/sub-section was found to be too general and not specific (can be combine with other sub-section) with the research title and objectives. Many sub-section and paragraph was found to be repetitive and redundant. It is preferable to have a sub-section with a heading directly discussing the objectives of the study.

 The structure of the manuscript is revised with additional sub-sections to improve the flow and easy understanding. Redundancy in the manuscript is avoided, authors have made significant changes in the manuscript for the materials, results and discussion sections with introduction of new sub-sections to improve readability of the manuscript. 

 Text in the manuscript also updated as follows: Sub-section “Crop data and model parameterization” (2.3) is further divided into sub-sub sections so that readers can understand efforts made by the authors in collecting detailed information for setting up crop simulation model to mimic real conditions. Section (3.6) “Adaptation strategies” is divided into sub-sections highlighting the proposed adaptation options that successfully offset the negative impacts of climate change. 

 I would strongly advice that a proper step by step procedure on the methodology involved should be prepare to clarify how the objectives was achieved. It was difficult to get a clear picture of the future projection (Tmin, Tmax and Prec) for the study because the authors are employed the 20 GCMs individually. The methodology can be improved by using an ensemble of GCMs, instead of using individual GCMs which introduce a lot of uncertainty in your final output. I would advise the authors to make an ensemble of the 20 GCMs or make an ensemble of the selected GCMs which can better modulate the local climate. Otherwise, the future projection output will have a big uncertainty, confusing and meaningless. The high uncertainty in the output, where some GCMs give positive results, while other GCMs give negative results, would render the results to meaningless as it seems unreliable to at least predict a certain degree of certainty in the future. It is understandable that different GCMs will have its own inherent variability. Therefore, an ensemble methodology should be used to reduce such uncertainty.

 We would like to submit that detailed step by step procedure followed which is explained in methods section, further we submit that efforts were made to collect detailed information on individual paraments before setting up the crop-simulations models in the study region. Authors have used 20 GCMs in the study as use of multiple models was suggested to provide more reliable assessment of impacts of climate change on weather sensitive sectors like agriculture (Asseng, 2013; Rosenzweig et al., 2013; Wilby et al., 2009). As suggested in earlier studies several scientists highlighted that cropping systems impact studies of climate change should not be assessed using only one GCM as major source of uncertainty for projections of future climate are from unknown future trajectories of CO2 and CH4, emissions, but also due to the highly simplified representation of reality encoded in these models (Wilby et al., 2004). Further the use of multiple models can provide more reliable decision support in climate change impact assessment and assessments of agricultural system vulnerability (Asseng, 2013; Rosenzweig et al., 2013; Wilby et al., 2009). We used 20 available climate models and two emission scenarios (RCP 4.5 and 8.5) in combination with multi-crop models to assess possible potential impacts of climate change on maize production. This study allows researches to understand the full spectrum of impacts of climate change on maize cropping system and further develop robust adaptation options to offset the negative impacts of climate change and at the same time maximize the productivity of maize systems in regions that are benefiting from climate change. Multi-climate model ensemble selection method has its strengths and weaknesses that often prove more trouble than they’re worth in addressing the impacts of climate change particularly on agricultural systems. Hence, we authors were convinced to use 20 GCMs, however in future studies we will try to use GCM ensemble in addition to 20 GCMS to understand its impact on crop yields

 It is not clear how the authors get the results for certain output, either through individual GCMs or taking a mean based on 20 GCMs (it should be ensemble, not mean). Due to lack of clarity in the methodology involved in getting the final output of certain section/sub-section, the authors seem to be confused about what it means by GCM ‘ensemble’.

 In the present study possible impacts of climate on each AEZs/soil type/maize cultivar/management options deployed are assessed independently. Results presented here are the deviations of maize yields in the future compared to baseline simulations. In order to show the impacts on the maize systems and the factors which are most vulnerable are computed by calculating the mean yield deviations. This is clearly explained in the materials section. 

 would be a major change in the results/output of the research if a proper ensemble method or any method selecting the best GCMs was applied, instead of using 20 individual GCMs (which introduce high uncertainties). I would advise for major revision of the article at its current state because without improve methodology, there will be a high uncertainty in the result/output of this article. Other specific comments can be found below.

 A major revision is performed to improve the manuscript, as mentioned earlier, we have completely revised the manuscript to incorporate the changes suggested by reviewers. After careful evaluation and discussions with global AgMIP leaders on the limitations of multi-climate ensemble for agricultural systems and we have highlighted in the materials and discussion sections why we have used 20 GCMs for the current study. However, we will definitely in future include an ensemble mean of GCMs in all our further studies to understand how the ensemble is impacting the crop yields

Specific comments

Abstract:

 Line 9: Website link address can be insert at the Materials and Methods section

 As suggested by the reviewer’s web link is removed for the abstract

 Line 11: It then develops an adaptation strategy that is locally > Then, adaptation strategy was develop that is locally...

 Changes are incorporated as suggested 

 Line 12: that help offset > to offset

 Changed as suggested

 Line 15: but differed in the magnitude of that increase > with different magnitude

 Changed as suggested

 Line 16: higher and those by CanESM2 > higher than CanESM2, INMCM4 and NorESM1-M under both emission scenarios.

 Changed as suggested

 Line 18: abbreviations for minimum temperature (Tmin) and maximum temperature (Tmax) can be use here. Please change for the rest of the manuscript.

 As suggested Tmin and Tmax are used in the entire manuscript

 Line 24-27: This sentence is too long to be in the abstract and confusing. What is the meaning of 'short duration varieties' and 'long duration varieties'? Do you mean ‘short duration maize varieties’…?

 Completely re-written, short-duration/ long-duration maize cultivars here refer to the time taken from sowing to crop maturity. In general, short-duration maize cultivars normally take 90 days to mature while, long-duration maize cultivars take 110 to 130 days to complete life-cycle.

 Line 34: at national and local levels and make their promotion easier > at national and local level.

 Changed as per the suggestion

 Line 35-38: This sentence is too long. Need to be rephrase and simplify.

 Sentence is completely revised and shorten

Introduction:

 Line 57: plant available water? Is this a scientific term?

 Plant available water (AW) is a technical term: defined as the difference between field capacity, FC, and wilting point, WP. The formula is:

AW=FC-WP

 Line 107: are viable is required > are viable.

 Changed as suggested

 Line 122: to adapt to the same > for adaptation

 Changed as suggested

 Line 124-126: Developing downscaled location specific climate change scenarios > Downscaling site-specific climate change scenarios for various agro-ecological zones (AEZs) of Embu county in Kenya under RCP4.5 and RCP8.5 at the mid and end century period.

 Changed as suggested

 Line 127: on the performance of maize > on maize yields

 Changed as suggested

 Line 128: and to identify key vulnerabilities? What do you mean here? To identify vulnerable area or to identify vulnerability factor?

 Objective is revised as: “Assessing the impacts of climate variability and change on maize yields in different AEZs of Embu county and identify key vulnerabilities to climate factors”. 

 Line 129: the objective is for a general agricultural system or specific to maize agricultural system?

 The objective is particularly focused for maize cropping system in eastern Kenya; however, this methodology remains same for other crops with different adaptation options

Materials and Methods:

Study Location:

 Line 157: potential area for what? agriculture? which crop?

 Low/High potential here refers to agricultural crops, sentence is revised to represent maize system in this context.

 Line 161: do you mean long duration maize varieties?

 In general, short-duration maize cultivars normally take 90 days to mature while, long-duration maize cultivars take 110 to 130 days to complete life-cycle. In the entire manuscript maize varieties are replaced with maize cultivars to avoid any confusion.

Crop data and model parameterization:

 Line 241: required data > the required data

 Changed as suggested

 Line 248: and the same were derived from the data collected > and the same data was collected...

 Changed as suggested

 Line 249: were derived from the amount > were based on the amount

 Changed as suggested

 Line 265: NO3 and NH4 > NO3 and NH4

 Changed as suggested, care has taken to represent the subscript for chemical formulas in the entire manuscript

 Line 267: 50 mm and 0.75 m

 The sentence is revised and both depth and plant spacing are represent with same unit (cm)

Results:

Observed trends in climate:

 Line 278: also in the temperatures > also in the precipitation

 Revised as suggested

Projected changes in future climate:

 Line 326: for the crops grown in that AEZ > for the crops grown in the study area.

 Revised as suggested

 Line 341: I think there is a major difference as well between SR and LR season under RCP4.5 although the difference under RCP8.5 is bigger.

 Revised as suggested

Impacts of climate change on maize yields:

 Line 389: to mid and end century > at the mid and end century

 Revised as suggested

 Line 403-405: This sentence is redundant. Please remove.

 Removed as suggested

 Line 406-412: It is better to have an ensemble of future projection to reduce the uncertainty rather than using and discussing each individual GCMs which could entail lengthy and contradictory results.

 Several researchers have made a case for seeking consensus amongst using multi-climate ensemble and individual GCMs projections. Agricultural systems responses to climate change is not linear to projected changes in surface temperatures or rainfall, it is a complex interaction where climate along with cultivar, soil and crop management options have significant impact on growth, development and finally productivity. In this study we used multi-climate models (20 GCMs) to assess the possible impacts of climate change to cover broad spectrum of impacts so that location specific adaptation options can be developed to cover wide range of impacts.

 Line 415-: How did you get the exact percentage result here? Previously, you discuss that there is a range of output by different 20 GCMs being used. You should not take the mean of the output from each GCMs here. Only an ensemble can do.

 Crop model simulated future maize yields deviation from baseline (seasons independently) are computed for each GCM. The average deviation is reported here. In the revised manuscript the range of deviation along with mean deviation are provided to avoid any confusion. Studies using multi-climate models particularly, agricultural systems the impact is often represent as mean deviation from baseline unlike meteorological studies where projections are characterized using multi-climate model ensembles. Few studies used multi-climate model ensembles, however, they failed to represent the full range of possible impacts of climate change on agricultural systems and the strategic adaptation options developed failed to cover all the possible climate projections. 

Effect of management on climate change impacts on maize yields:

 Line 429: the effect of variety > the effect of maize variety

 Revised as suggested

 Line 434-: The different and contradictory result might be due to the usage of 20 GCMs individually, instead of using an ensemble of GCMs. This might be the source of high uncertainty in your output by APSIM and DSSAT.

 Crop simulations models APSIM and DSSAT have structural differences in simulation function. Large variation among the crop models because of different assumptions for parameter functions [Boote et al., 2013] like cardinal temperature (CR), photoperiod and low temperature sensitivity. As a result, both the crop simulation models marginally differed in assessing the impacts of climate change on maize cultivars.

 Line 442: simulation analysis > simulation analysis,

 Revised as suggested

 Line 477: What does it means by early planted maize? Can you specify the exact date and duration here?

 Early planting maize here refers to early sowing relative to normal sowing date. Since the planting window changes for each season and AEZs the details are not provided in the text. However, the changes in planting dates (early planting) for each season and AEZ are provided in table 7 and which is referred in the text.

Impacts under elevated carbon dioxide levels

 Line 493: It is better to have an ensemble of GCMs, rather than individual GCMs result as it will introduce high model uncertainty in your DSSAT and APSIM modelled yields.

 As explained earlier, the study assesses the combined effects of climate change, soil, maize cultivar, AEZ, CO2 fertilization and management options on future maize systems to cover full range of possible impacts. This allows policy makes and stakeholders to understand wide range of impacts under climate scenario and there by develop adaptations that considers all possible impacts on maize systems.

Adaptation strategies:

 Line 510: performing variety > performing maize variety

 Revised as suggested and variety is replaced with cultivar in the entire manuscript

 Line 535: which propose intervention do you mean here?

 Here “proposed interventions” refers to the adaption strategies such as changing planting dates, plant population, fertilizer amounts and maize cultivars. The whole section is revised and sub-sections are introduced to highlight the adaptation strategies that were recommended in this manuscript for easy understanding to readers.

Discussion:

 The first paragraph is just a general statement and can be remove or relevantly insert at various part of other section/sub-section.

 Whole discussing is revised and we would like to thanks the reviewers for the critical comments that helped in improving the manuscript. As suggested the first paragraph is removed 

 Line 552: effectiveness > the effectiveness

 Changes incorporated as suggested

Climate variability and change

 This sub-section is too general and doesn't reflect with the research title and doesn't reflect with the research objectives on maize yields. It is better to directly integrate the results being obtained here and relate it with other relevant sub-section which reflect your main findings (related to maize yield).

 The section is completely revised as per the suggestions

 The first paragraph is redundant. It is better to directly discuss your results.

 Revised as per the suggestions and removed repeated text

 The second paragraph should be discuss in data section

 Revised and moved as per the suggestion

 Line 594: Daily Temperature Range (DTR) > Diurnal Temperature Rang (DTR)

 Revised and changed Daily to Diurnal

 Line 611: 'we used an ensemble of 20 GCMs' > This statement is confusing with your previous methodology and discussion. Are you using GCMs individually or an ensemble of GCMs? The author should have a proper understanding of what 'ensemble' means.

 Revised the complete section to avoid the confusion on ensemble, in this study multi-climate model ensemble is not used rather 20 GCMs were used that are able to represent broadly from the distribution of projections.

 Line 612-613: This statement should be discuss in the methods and data section. The reason for selecting only two different emission scenario (RCP4.5 and RCP8.5) and two time period (mid and end century) should be properly justify.

 As per the suggestion this is moved to material section and justified why the study uses RCP 4.5 and 8.5.

 Line 615-: This section should be in the methods section. It is repetitive.

 Repetitive text in the manuscript are removed and as per the suggestions from reviewers it is moved to methods section

Impacts of climate variability and change on maize production

 The statement in the first paragraph is general making it lengthy. It should directly discuss the sub-section topic on 'Impacts of climate variability and change on maize production'

 Revised and general text is removed as per the suggestion

Conclusions:

 The first paragraph is redundant and repetitive from previous section. It should be remove.

 Removed and the whole section is revised

 Line 745: This is the first time that 'multi-model ensemble' being mention, and it was mention at the conclusion. This is confusing. Your results discuss individual GCMs, not an ensemble of GCMs. Please clarify.

 Revised the whole sections and removed multi-climate ensemble to avoid any confusion to readers

 Line 763-: The specific discussion and specific results for the intervention (changing variety, plant density, soil fertility management etc.) are scatter across the article and not properly and directly discuss in sub-section ‘Adaptation strategies’. It is difficult to understand how these interventions play a key role in maize production in relation with climate change impact.

 In the discussion section particularly, ‘Adaptation strategies’ is completely revised and sub-sections were included to describe how the proposed interventions influenced to offset the negative impacts of climate change and at the same time in regions where the projected impacts are beneficial to maize systems, adaptation options maximized maize yields even in low input systems 

Reviewer #2: (PONE-D-20-14901) Simulating possible Impacts of climate variability and change on Maize production in Embu County, Kenya.

Abstract:

 L9: Eliminate “(www.agmip.org)”.

 Removed web link as suggested

 There are mixed verb tenses, somewhere use the present tense “L8: The study assesses…..”, and other places use the past tense, “L12: The study used…”. Please use the present or past tense in the whole of the paper.

 Carefully revised the entire manuscript to avoid mixed verb tenses

 Please rewrite lines 13 to 15.

 Revised the lines between 13 to 15 

 It would be better to highlight your contribution, more clearly in abstract.

 Revised abstract to highlight our recommended adaptions we proposed 

Introduction:

 The authors should highlight the research significant, more clearly in the introduction section.

 Revised introduction as suggested to highlight significance of the current research

 There are some grammatical errors in this section. Please check whole the section carefully.

 Authors and AgMIP leaders improved the manuscript including grammatical errors

 References are not always relevant and up-to-date. This reviewer suggests the inclusion of a few relevant publications. The list is given below:

https://rmets.onlinelibrary.wiley.com/doi/abs/10.1002/joc.6307

https://link.springer.com/article/10.1007/s11368-020-02632-0

https://link.springer.com/article/10.1007/s00704-019-02979-6

https://www.sciencedirect.com/science/article/pii/S0168192319302199

 References provided are used where ever they are applicable

Method:

 It is mentioned in L133 that the Embu County in Kenya is adopted as the case study. What are other feasible alternatives? What are the advantages of adopting this particular case study over others in this case? How will this affect the results? The authors should provide more details on this

 The study selected Embu county for the following:

 Embu county is one of the potential agricultural regions characterised by a range of agro-ecological zones ranging from highlands to lowlands.

 The county was selected based on its representativeness of the country’s major agro-ecological zones and based on the availability of the data (crop, soil and climate) required to parameterize the crop simulation models.

 The details are provided in the section 2.1 “Study location”

 It is mentioned in L181 that historical records of 1980 to 2010 are taken. Why are more recent data not included in the study? Is there any difficulty in obtaining more recent data? Are there any changes to the situation in recent years? What are its effects on the result?

 Efforts were made to collect latest data, however, due to limitations in sharing climate records as per the country policy with CGIAR institutes this study only considered to represent baseline between 1980-2010 

Results and discussion

 From Figure 6, it is evident that the authors obtained poor performance (R2: 0.33-0.44) during the validation phase. Is it a reliable model for crop simulating?

 The poor performance of the crop simulation models in reproducing the maize yields for the year 2013 are due to large number of factors which are discussed in detail in the result section (3.3 Crop model calibration and validation). Crop simulation model’s performance in reproducing experiment yields are in good agreement while in the farmers fields the poor correlation is due to number factors such as differences in interpreting and translating farmer description of the resource endowment into model parameters, inability of the models to capture the effects of biotic stresses such as pests, diseases and weeds, inaccuracies in estimating per hectare yields from bags per plot as reported by farmers and inaccuracies in defining the initial conditions. However, the simulated long-term yields of different AEZs reflected the trends in the yields reported by farmers fairly well, especially in the low potential LM4 and LM5 AEZs. In these AEZs, high moisture stress is the major yield limiting factor and this to a large extent masks the relatively low effect of other management practices and also the influence of differences in the resource base.

Conclusion:

 In the conclusion section, the limitations of this study suggested improvements in this work, and future directions should be highlighted.

 The limitations of the study and further improvements required are described in the conclusions as suggested

---

## [Decision Letter · Decision Letter 1]

8 Sep 2020

PONE-D-20-14901R1

Simulating adaptation strategies to offset potential impacts of climate variability and change on maize yields in Embu County, Kenya

PLOS ONE

Dear Dr. Gummadi,

Thank you for submitting your manuscript to PLOS ONE. After careful consideration, we feel that it has merit but does not fully meet PLOS ONE’s publication criteria as it currently stands. Therefore, we invite you to submit a revised version of the manuscript that addresses the points raised during the review process.

We look forward to receiving your revised manuscript.

Kind regards,

Shamsuddin Shahid

Academic Editor

PLOS ONE

Additional Editor Comments (if provided):

Reviewer 1 pointed out several specific issues. Authors are requested to revise the article based on the comments.

Reviewers' comments:

Reviewer's Responses to Questions

**Comments to the Author**

1. If the authors have adequately addressed your comments raised in a previous round of review and you feel that this manuscript is now acceptable for publication, you may indicate that here to bypass the “Comments to the Author” section, enter your conflict of interest statement in the “Confidential to Editor” section, and submit your "Accept" recommendation.

Reviewer #1: All comments have been addressed

Reviewer #2: All comments have been addressed

2. Is the manuscript technically sound, and do the data support the conclusions?

Reviewer #1: Yes

Reviewer #2: Yes

3. Has the statistical analysis been performed appropriately and rigorously? 

Reviewer #1: No

Reviewer #2: Yes

4. Have the authors made all data underlying the findings in their manuscript fully available?

Reviewer #1: Yes

Reviewer #2: No

5. Is the manuscript presented in an intelligible fashion and written in standard English?

Reviewer #1: Yes

Reviewer #2: Yes

6. Review Comments to the Author

Reviewer #1: General Comments

The manuscript has been greatly improved by the authors after the first revision. However, there is still few improvements on the arrangement of the section/sub-section that can be done (refer to the specific comments). In term of methodology, although the authors have reply properly the comments given, some of it wasn’t inserted into the manuscript (refer to the specific comments). The methods and results are still mixed-up in some of the section/sub-section which the authors can make in a more proper order and sequence. Some clarification on how the future changes was calculated has been reply through peer-review comments but wasn’t mentioned anywhere in the manuscript. There are some minor but substantial details on the results and discussion that need to be revised.

Other specific comments can be found below.

Specific comments

Abstract:

•Line 19: There are 20 GCMs used. Therefore, projection by HadGEM2-CC, HadGEM2-ES, and MIROC-ESM should be higher than the rest of 17 GCMs. Why is it mention here, projection of HadGEM2-CC, HadGEM2-ES, and MIROC-ESM is higher than CanESM2, INMCM4 and NorESM1-M only?

•Line 31: Are you 'develop' the adaptation practices or 'incorporate' the adaptation practices into the model?

•Line 33: I think you are incorporating the operational adaptation strategies using the readily available technologies here, not develop the operational adaptation strategies using the readily available technologies.

Section 1. Introduction

•Line 63: which are > which have a marginal

•Line 63: adding to this > adding to this,

•Line 125: Instead of using the term climate, a more specific term, rainfall, Tmin, Tmax (climate indices that you assess) should be mention here. Climate can be a lot of things.

•Line 127: Repeated abbreviation for AEZs

•Line 127: various AEZs, namely,... > the zones used in the study should be mentioned here.

•Line 130: Are you developing adaptation options?

Section 2.1 Study Location

•Line 141: rising > up to

•Line 157: AEZs > AEZs of

•Line 163: higher use > higher application

2.2 Assessing current climate variability and future climate conditions

•Line 166: Assessing? Are you making any assessment here? There is no discussion on the assessment of current climate variability in this sub-section. The heading is confusing.

•Line 181: Please be notify that the font size is different here. Please check throughout the manuscript.

•Line 190: pathway are > pathway

•Line 193-196: The equation number for equation (1)-(4) should be align right and the equation should be centre.

Section 2.3

•Line 234: I don't think you need to give this heading. Sub-heading 2.31 can be change to sub-heading 2.3

2.3.1 Crop Simulation Models (CSMs)

•Line 241: were used to assess > to assess

•Line 246: Please recheck if there is any more repetitive abbreviation throughout the manuscript.

2.3.2 Model parameterization

•255: Sub-heading 2.3.2 can be change to 2.4. While sub-section Farm and farm management data can be change to 2.4.1 etc for the rest of sub-heading.

Farm and farm management data:

•Line 271-276: Please re-phrase this sentence.

•Line 281: During In the survey > During the survey

•Line 283: This formed?

Model calibration and validation:

•Line 312-323: This discussion should be put into results or discussion section.

Crop simulations:

•Line 332: NO3, NH4 > NO3, NH4

•Line 337: and or > and

•Line 342: crop simulation model > CSM

3. Results

•There should be a sub-section under Results to discuss the performance of APSIM and DSSAT in term of calibration and validation of the model.

3.1 Trends in observed climatic conditions

•Line 359: How do you measure the trend? Simple Linear Regression?

•Line 360: delete 'and in the rate of increase'

•Line 361: Why are you comparing Tmax and Tmin for period 2001-2010 with period 1981-1990? You can make a simple regression line to see the increasing rate of Tmin/Tmax from the whole historical period of 1981-2010.

•Line 367: If you are using simple linear regression for trend, you can give the increasing/decreasing rate here.

•Line 373-378: You discuss the results in the previous paragraph. Only now, you discuss the method that you use. This is not in proper order. Also, please give briefly the method that you use for trend analysis in the method section. Not in results section.

3.2 Projected changes in climate conditions

•Line 380: surface temperature > use directly the specific temperature indice that you employ, Tmin and Tmax. Surface temperature can be a lot of things.

•Line 385: You are employing 20 GCMs, but these 4 GCMs (HadGEM2-CC, HadGEM2-ES, IPSL-CM5A-MR, IPSL-CM5A-LR) is higher compare to only other 4 GCMs (CanESM2, INMCM4, MRI-CGCM3, and NorESM1-M)? This is confusing.

•Line 388: Projected increase was being compare with which historical period?

•Line 391: Please use a consistent decimal places for temperature throughout the manuscript.

•Line 397: 3.5 oC > 3.5oC, make similar correction throughout the manuscript

•Line 398: Give numerical value for the result and discussion. What is the value for current temperature? What is value for low temperature event? What is value for high temperature event?

•Line 400: sift > shift

Impacts across AEZs:

•This can be sub-heading 3.2.1. Do the same for the rest.

Impacts across seasons:

•Line 447: delete 'DSSAT'

•Line 451: what is the numerical value for baseline yields?

Impacts of plant population:

•Line 496: population > plant population

Impacts of soil fertility and nitrogen application:

•Line 502: How exactly climate change give positive impacts? Increase rainfall/temperature?

3.3 Adaptation strategies

•Line 527: What exactly is adapted technology that you mentioned here?

•Line 531: All of this results discussing about adapted technology? or certain management practice? Please make it clear. In the first paragraph of this section, you are talking about various management practice, then you are talking about adapted technology. Now, the results are talking about which one? This is confusing.

4.1 Assessing variability in the current and projected climatic conditions

•Section 4.1 > This heading is unnecessary

4.1.1 Climate Variability

•Line 567: variability in term of temporal or spatial?

4.2 Impacts of climate variability and change on maize production

•Line 644: was done carried out > was carried out

•Line 659: AEZs > AEZs of

•Line 660: AEZs > AEZs of

4.3 Adapting to climate change

•Line 705: This sub-section seems to be a general statement. May be can briefly incorporate in the conclusion.

Figure 2

•The outside borderline box should be remove. Do the same for all figure except fIg. 1 as the borderline box is require for lat/lon information.

Figure 5

•The Legend can be improve

•Please re-check the font size of the caption

Figure 7

•Please put dot line between AEZs. The series overlap can be made to 0%. Do the same for Fig. 9 as well.

Figure 8

•Series overlap can be made to 0% for each GCM

Table 1

•Table can be improved. No color shading should be use and a proper bordering line should be employ. Please refer to other research article on how to prepare a proper table for research publication. Do the same for all Table.

Reviewer #2: The authors have addressed all of the my comments. So, I suggest to publish the paper in current form.

7. PLOS authors have the option to publish the peer review history of their article (what does this mean?). If published, this will include your full peer review and any attached files.

Reviewer #1: **Yes: **Zulfaqar Sa'adi

Reviewer #2: No

---

## [Author Response · Author response to Decision Letter 1]

23 Sep 2020

Dear Reviewers, 

We truly appreciate the thorough review and comments offered on our article. After careful consideration of the general and specific comments suggested, we have drafted our responses as attached and made appropriate changes in the revised article. we author’s thanks the reviewers for their positive comments for the improvements made on our manuscript. The structure of the manuscript is revised as per the suggestions of the reviewer, particularly on the rearrangement of section/sub-sections. Authors have made changes in the manuscript for the for all the sections as suggested.

Reviewer #1: (PONE-D-20-14901) Simulating possible Impacts of climate variability and change on Maize production in Embu County, Kenya.

---

## [Decision Letter · Decision Letter 2]

1 Oct 2020

PONE-D-20-14901R2

Simulating adaptation strategies to offset potential impacts of climate variability and change on maize yields in Embu County, Kenya

PLOS ONE

Dear Dr. Gummadi,

Thank you for submitting your manuscript to PLOS ONE. After careful consideration, we feel that it has merit but does not fully meet PLOS ONE’s publication criteria as it currently stands. Therefore, we invite you to submit a revised version of the manuscript that addresses the points raised during the review process.

We look forward to receiving your revised manuscript.

Kind regards,

Shamsuddin Shahid

Academic Editor

PLOS ONE

Reviewers' comments:

Reviewer's Responses to Questions

**Comments to the Author**

1. If the authors have adequately addressed your comments raised in a previous round of review and you feel that this manuscript is now acceptable for publication, you may indicate that here to bypass the “Comments to the Author” section, enter your conflict of interest statement in the “Confidential to Editor” section, and submit your "Accept" recommendation.

Reviewer #1: All comments have been addressed

2. Is the manuscript technically sound, and do the data support the conclusions?

Reviewer #1: Yes

3. Has the statistical analysis been performed appropriately and rigorously? 

Reviewer #1: Yes

4. Have the authors made all data underlying the findings in their manuscript fully available?

Reviewer #1: No

5. Is the manuscript presented in an intelligible fashion and written in standard English?

Reviewer #1: Yes

6. Review Comments to the Author

Reviewer #1: General Comments

The manuscript has been greatly improved by the authors since the first revision. The manuscript section/sub-section has been arranged in a good order. There are some minor revision needed, given in the specific comments below.

Specific comments

Abstract:

•Line 20: than rest > than the rest

1.Introduction

•Line 78: projected climate conditions > projected climate conditions on crop growth

•Line 126: maximum and minimum temperatures > Tmax and Tmin

2.2 Current climate variability and future climate conditions

•Line 172: All of the four stations > All stations

•Line 198: Representative Concentration Pathways (RCPs) > RCPs

•Line 200: W m−2 > W m-2

•Line 209: Arrange the equation properly

•Line 227: as use > the usage

•Line 234: one GCM, major > one GCM, as major

2.4.1 Farm and farm management data:

•Line 264: Delete the colon (:). Do the same for the rest.

2.4.2 Soil data

•Line 283: form > from

•Table 2: What is soil type for Machanga? Please explain briefly the missing soil parameter in the discussion.

2.4.4 Model calibration and validation

•Line 318: CSMs s were > CSMs were

•Line 318-319: Please rephrase the sentence.

•Line 324: Table 4 > Table 4.

•Line 324: of 160 > from 160

•Line 328: are generally > are found to be generally

2.4.5 Crop simulations

•Line 353: left over > leftover

3.1 Trends in observed climatic conditions

•Line 403: expected have > expected to have

3.2 Projected 406 changes in climate conditions

•Line 413: compared to to other > compared to other

•Line 446: What is this sentence? "Impacts of climate change on maize productivity of the impact on maize." Is this a new sub-heading?

4.1 Climate Variability

•Line 631: log-term > long-term

4.3 Impacts of climate variability and change on maize production

•Line 714: and by when > and when

•Line 712-716: The sentences is too long. It can be separated into two sentences.

7. PLOS authors have the option to publish the peer review history of their article (what does this mean?). If published, this will include your full peer review and any attached files.

Reviewer #1: **Yes: **Zulfaqar Sa'adi

---

## [Author Response · Author response to Decision Letter 2]

7 Oct 2020

Dear reviewer,

We truly appreciate the thorough review and comments offered on our article. After careful consideration of the general and specific comments suggested, we have drafted our responses as shown below and made appropriate changes in the attached revised article 

Reviewer #1: (PONE-D-20-14901) Simulating possible Impacts of climate variability and change on Maize production in Embu County, Kenya.

General Comments

The manuscript has been greatly improved by the authors since the first revision. The manuscript section/sub-section has been arranged in a good order. There are some minor revision needed, given in the specific comments below. 

• As suggested by the reviewer, minor changes are incorporated in the manuscript

Specific comments

Abstract:

• Line 20: than rest > than the rest

Changed as suggested

1. Introduction

• Line 78: projected climate conditions > projected climate conditions on crop growth

Changed as suggested

• Line 126: maximum and minimum temperatures > Tmax and Tmin

Changed as suggested

2.2 Current climate variability and future climate conditions

• Line 172: All of the four stations > All stations

• Changed as suggested

• Line 198: Representative Concentration Pathways (RCPs) > RCPs

• Changed as suggested

• Line 200: W m−2 > W m-2

Changed as suggested

• Line 209: Arrange the equation properly

Changed as suggested

• Line 227: as use > the usage

Changed as suggested

• Line 234: one GCM, major > one GCM, as major

Changed as suggested

2.4.1 Farm and farm management data:

• Line 264: Delete the colon (:). Do the same for the rest.

Changed as suggested and in rest of the sub-sections, colon (:) is deleted

2.4.2 Soil data

• Line 283: form > from

Changed as suggested

• Table 2: What is soil type for Machanga? Please explain briefly the missing soil parameter in the discussion. 

Soil Type for Machanga is “Xanthic Ferralsol” and the missing soil texture details are incorporated in the table as suggested

2.4.4 Model calibration and validation

• Line 318: CSMs s were > CSMs were

Whole sentence is revised

• Line 318-319: Please rephrase the sentence.

• Whole sentence is revised as instructed

• Line 324: Table 4 > Table 4.

Changed as suggested

• Line 324: of 160 > from 160

Changed as suggested

• Line 328: are generally > are found to be generally

Changed as suggested

2.4.5 Crop simulations

• Line 353: left over > leftover

Changed as suggested

3.1 Trends in observed climatic conditions

• Line 403: expected have > expected to have

Changed as suggested

3.2 Projected 406 changes in climate conditions

• Line 413: compared to to other > compared to other

Changed as suggested

• Line 446: What is this sentence? "Impacts of climate change on maize productivity of the impact on maize." Is this a new sub-heading?

Sub-heading in the previous version, this is now deleted from the manuscript

4.1 Climate Variability

• Line 631: log-term > long-term

Changed as suggested

4.3 Impacts of climate variability and change on maize production

• Line 714: and by when > and when

Revised the entire sentence

• Line 712-716: The sentences is too long. It can be separated into two sentences.

As suggested by the reviewer, the whole sentence is revised and separated into two small sentences.

---

## [Editor Report · Decision Letter 3]

9 Oct 2020

Simulating adaptation strategies to offset potential impacts of climate variability and change on maize yields in Embu County, Kenya

PONE-D-20-14901R3

Dear Dr. Gummadi,

We’re pleased to inform you that your manuscript has been judged scientifically suitable for publication and will be formally accepted for publication once it meets all outstanding technical requirements.

Kind regards,

Shamsuddin Shahid

Academic Editor

PLOS ONE
---

## [Editor Report · Acceptance letter]

16 Oct 2020

PONE-D-20-14901R3 

Simulating adaptation strategies to offset potential impacts of climate variability and change on maize yields in Embu County, Kenya 

Dear Dr. Gummadi:

I'm pleased to inform you that your manuscript has been deemed suitable for publication in PLOS ONE. Congratulations! Your manuscript is now with our production department. 

Kind regards, 

on behalf of

Dr. Shamsuddin Shahid 

Academic Editor

PLOS ONE